# Two Tickets are Better than One:
# Fair and Accurate Hiring Under Strategic LLM Manipulations

**Lee Cohen** [* 1]   **Connie Hong** [* 1]   **Jack Hsieh** [* 1]   **Judy Hanwen Shen** [* 1]

## Abstract

In an era of increasingly capable foundation models, job seekers are turning to generative AI tools to enhance their application materials. However, unequal access to and knowledge about generative AI tools can harm both employers and candidates by reducing the accuracy of hiring decisions and giving some candidates an unfair advantage. To address these challenges, we introduce a new variant of the strategic classification framework tailored to manipulations performed using large language models, accommodating varying levels of manipulations and stochastic outcomes. We propose a "two-ticket" scheme, where the hiring algorithm applies an additional manipulation to each submitted resume and considers this manipulated version together with the original submitted resume. We establish theoretical guarantees for this scheme, showing improvements for both the fairness and accuracy of hiring decisions when the true positive rate is maximized subject to a no false positives constraint. We further generalize this approach to an $n$-ticket scheme and prove that hiring outcomes converge to a fixed, group-independent decision, eliminating disparities arising from differential LLM access. Finally, we empirically validate our framework and the performance of our two-ticket scheme on real resumes using an open-source resume screening tool.

## 1. Introduction

Hiring decisions can profoundly impact an individual's professional path and long-term success. As algorithmic tools are increasingly deployed to recommend or make these decisions, they have rightfully come under scrutiny from economists (Hu & Chen, 2018; van den Broek et al., 2020),

*Equal contribution [1]Stanford University, Palo Alto, CA, USA. Correspondence to: Judy Hanwen Shen <jhshen@stanford.edu>.

*Proceedings of the 42$^{nd}$ International Conference on Machine Learning*, Vancouver, Canada. PMLR 267, 2025. Copyright 2025 by the author(s).

journalists (Lytton, 2024), and policy makers (City of New York, 2021). AI tools that exhibit undue biases and unexplainable behavior present a major barrier to achieving accountability in these algorithmic hiring schemes (Dastin, 2018). Although algorithmic hiring tools are designed with the goal of hiring the best candidates, these tools may not be robust to candidates manipulating their application materials. For example, Applicant Tracking Systems (ATS) are widely used to screen qualified candidates in the first stage of the hiring process: however, there exist many tutorials for manipulating resumes for this system (Stahl, 2022). This problem has been studied through the lens of *strategic classification* where individuals can manipulate their inputs (e.g., a job application) to influence the decision made by a classifier (e.g., a hiring algorithm) (Hardt et al., 2016a; Kleinberg & Raghavan, 2020; Levanon & Rosenfeld, 2021). The goal of the hiring side is to design a strategy-proof selection algorithm, whereas the goal of the applicants is to maximize their utility: the difference between the benefit of receiving a positive prediction and the cost of manipulation. The applicant's "best response" is then the optimal way an individual should manipulate their input — given knowledge of the classifier's behavior — in order to maximize their utility. The challenge lies in designing classifiers that are robust to such manipulations while maintaining fairness and accuracy.

With the recent proliferation of generative AI services that are now widely used by job seekers (Chamorro-Premuzic, 2024), a new variable has been introduced to the algorithm hiring cycle and strategic classification. Writing or editing a resume using generative AI has become accessible and widespread. In a recent survey, 57% of respondents admitted to using AI to create their resume (Business Wire, 2025). Since candidates have no knowledge of how employers make hiring decisions, the optimal strategic classification response becomes straightforward: candidates edit their resumes using their preferred AI tool, opting for a premium version if they recognize its advantages and can afford it. As a result, those accessing better models may gain an unfair advantage in the selection stage of automatic hiring algorithms. Thus, companies may be filtering for candidates who used the best LLMs rather than candidates who are the most qualified.

The interactions between hiring algorithms and application-enhancing generative AI tools create a unique setting to examine fairness and strategic behavior. Since manipulation in this setting is low-effort, many candidates will choose to manipulate their resumes, even if this does not guarantee a positive outcome. This contrasts with the classic model, where individuals manipulate their input only when a positive outcome is achieved. Moreover, strategic classification in the era of LLMs introduces two key challenges: (1) unlike prior group-based fairness settings, the hirer cannot directly determine whether an application has been manipulated or which LLM was used, and (2) unlike the classic strategic classification setting, manipulations are stochastic, as LLM outputs are inherently non-deterministic. Motivated by this complex yet realistic interaction between strategically generated application materials and algorithmic hiring algorithms, our work presents a first step into modeling and analyzing algorithmic hiring ecosystems in the era of generative AI; our contributions are as follows:

- We show that some models enhance resume relevance scores more than other models, and that the benefits of repeated LLM manipulations stagnate (Section 3).

- We translate the empirical behavior of LLMs used for job applications into a realistic model for strategic classification (Section 4).

- We prove that under existing hiring schemes, disparities in access to LLMs lead to disparities in hiring outcomes, even under *stochastic* manipulations (Section 5) and an unknown deployed model.

- We introduce a two-ticket scheme where the hiring algorithm applies an additional LLM manipulation to each submitted resume and considers this manipulated version together with the original submitted resume. We prove that this scheme improves disparities among candidates and accuracy for employers. We also generalize the two-ticket scheme to an $n$-ticket scheme, proving that the $n$-ticket scheme eliminates group-dependent disparities as $n \to \infty$, with outcomes converging exponentially to a fixed, group-independent decision (Section 6).

- We validate our theoretical model and results through a case study using real resumes and an open-source resume scoring algorithm (Section 7), demonstrating that our two-ticket scheme enhances both fairness and accuracy in practice.

## 2. Related Work

**Strategic Classification**  Hardt et al. (2016a) introduce strategic classification as a Stackelberg game to address the

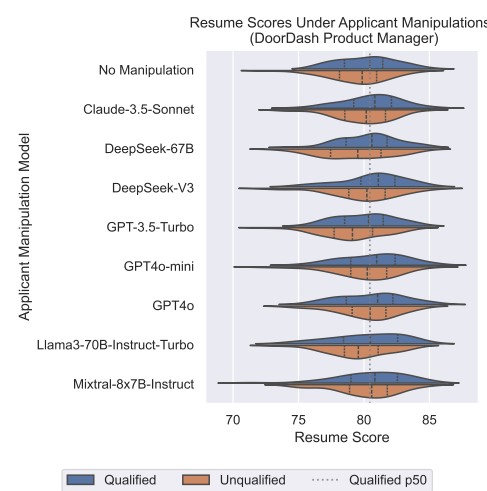

Figure 1. Resume score distribution of 50 qualified (matching occupation) and 50 unqualified (different occupation) resumes before and after LLM manipulations for a DoorDash Product Manager job posting. Models such as GPT-4o, Claude-3.5-Sonnet, and DeepSeek-V3 generate a distribution of unqualified resumes that is indistinguishable from qualified resumes.

impact of manipulative tactics on classification problems. We draw on several later works that provide a modified strategic classification game that models disparities in manipulation abilities (Hu et al., 2019b; Milli et al., 2019; Chen et al., 2020; Diana et al., 2024). Furthermore, we use techniques from prior work (Braverman & Garg, 2020) to describe "random" classifiers in light of stochastic strategic manipulations. Similarly to several previous works (Ghalme et al., 2021; Cohen et al., 2024), we assume that the deployed classifier is unknown to the candidates. Finally, previous work (Somerstep et al., 2024) has explored the use of casual strategic classification to understand labor market dynamics: our work extends this by highlighting the effects of strategic behavior on employer and labor force outcomes.[1]

## 3. Empirical Motivation: Stochastic Resume Manipulation using LMMs

Since prior works in strategic classification focus on deterministic manipulations, we empirically motivate our theoretical model of stochastic LLM manipulations. We prompt a variety of models to improve technology sector resumes (Drushchak & Romanyshyn, 2024).[2] We used a general prompt to simulate a job applicant who is aiming to ap-

---

[1]Full related work including fairness in algorithmic hiring and the risks of generative models can be found in Appendix A.

[2]See Appendix D for prompt details and prompt analysis experiments to reduce hallucinations.

ply to multiple jobs with the same enhanced resume (e.g., via recruitment agency). The resumes were then evaluated against the target job descriptions through open source software that scores resumes against a designated job description to produce a *relevance score*. This type of simple scoring model, as a first filter for resumes, is widespread, with 98.4% of Fortune 500 companies using them within applicant tracking systems (Purcell, 2024). We identified three key behaviors of LLM manipulations: (1) LLM manipulations stochastically enhance resume scores (Figure 4a),(2) The effectiveness of LLM manipulations varies by model: newer, premium LLMs improve resume scores more (Figure 1), (3) Improvements from manipulations stagnate with repetition: applying the same LLM repeatedly to the same resume results in diminishing changes (Figure 4b).

Figure 1 illustrates that using a simple job-agnostic prompt with an input resume significantly improves the scores computed by a resume screening system. A second observation that motivates our study of disparities is the differential outcomes resulting from applying different LLMs to a candidate's resume. Figure 3 shows the post-manipulation resume scores across a broad set of models. Using the dotted lines as a reference for the median score of the original resumes of the qualified group, it is evident that applying different LLMs has different effects on the outcome relevance score. Distinguishing qualified and unqualified candidates is already a difficult task, but candidate manipulation makes it harder. Some models, particularly the higher cost-to-access models (e.g., CLAUDE-3.5-SONNET, GPT-4O) improved the resume scores of the unqualified resumes so that they were indistinguishable or better than the qualified resumes without LLM manipulations, while cheaper or free-to-access models (e.g., GPT-3.5-TURBO, MIXTRAL-8X7B-INSTRUCT) did not significantly improve scores on average of the unmanipulated resumes regardless of qualification.[3] By qualitatively inspecting the manipulated resumes, we found that LLMs yielding larger score improvements (e.g., GPT-4O) better adhered to the traditional elements of a resume while less effective models simply reorganized the input resume. For example, GPT-4O also added additional elements such as a resume summary and dedicated sections for educational history.[4] We also found that all of the newer, premium models (particularly CLAUDE-3.5-SONNET), increased the average resume similarity (embedding distance) (Figure 8).

Finally, we also observed that repeated manipulations did not significantly alter resumes. The first round of modifications typically standardized language and formatting

according to a conventional resume structure. However, a second round of manipulations did not deviate substantially from the first. This observation is also reflected by the similarity of the resume score distributions of the once and twice-manipulated resumes (Figure 4b).

Together, these three key observations regarding LLM manipulations — the potential for resume improvement, the differences in results among various LLM models, and the stagnation in changes from multiple iterations — motivate our proposed model strategic manipulation in Section 4.

## 4. Model

We represent each candidate as a triplet $(\boldsymbol{x}, g, y)$, where $\boldsymbol{x} \in \mathbb{R}^d$ represents the candidate's original (unmanipulated) resume features; $g \in \{P, U\}$ denotes the group membership, with $P$ indicating the privileged group and $U$ indicating the unprivileged group; and $y \in \{0, 1\}$ represents the true label, with $0$ indicating an unqualified candidate and $1$ indicating a qualified candidate. It is important to note that we do not require that $\boldsymbol{x}$ fully determines $y$.

Our model accommodates any combination of $d_1$ *fundamental* and $d_2$ *style* features in the feature space (i.e., $d = d_1 + d_2$). Style features $(x_1, x_2, \ldots, x_{d_1})$ refer to attributes about a resume's presentation such as writing quality, vocabulary, and grammar, whereas fundamental features $(c_1, c_2, \ldots, c_{d_2})$ refer to technical attributes such as programming skills, years of experience, or educational background.[5]

Overall, we express each candidate's resume features as an $d$-dimensional feature vector in $\mathbb{R}^d$:

$$\boldsymbol{x} = [x_1, x_2, \ldots, x_{d_1}, c_1, c_2, \ldots, c_{d_2}].$$

We model the candidate population as a joint distribution $\mathcal{D}$ over feature vectors, group memberships, and true labels. We define the random variable triplet $(\boldsymbol{X}, G, Y) \sim \mathcal{D}$ with $\boldsymbol{X} \in \mathbb{R}^d$, $G \in \{P, U\}$, and $Y \in \{0, 1\}$. Moreover, we assume that both groups have identical distributions over resume feature vectors, and that the true label is independent of group membership — that is, we assume that $\boldsymbol{X}$ and $G$ are independent and that $Y$ and $G$ are conditionally independent given $\boldsymbol{X}$. For our model to be appropriate, each group comprises a non-negligible proportion of the population: that is, $\mathbb{P}(G = P), \mathbb{P}(G = U) > 0$.

### 4.1. LLM Manipulation

We assume some candidates are manipulating using LLMs (Verma & Renjarla, 2024; Stahl, 2022). In what follows, we formalize our model for LLM manipulation of

---

[3]Model pricing rapidly changes for consumer platforms. Furthermore, not all models are available on consumer platforms. We include a cost analysis for API access to simulate third-party career services with tiered offerings in Table 7.

[4]Representative examples can be found in Appendix C.4.

[5]If all features are fundamental features ($d_1 = 0$), then the scenario reduces to traditional non-strategic classification.

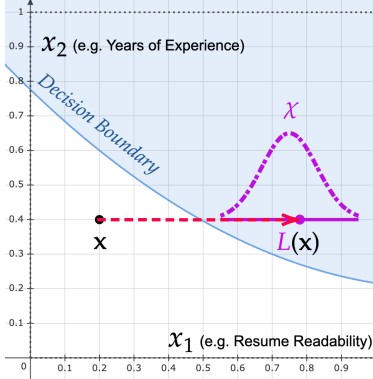

*Figure 2.* Visualization of an *LLM Manipulation* $L$ over $d = 2$ with one style feature ($x_1$) and one fundamental feature ($x_2$). Modifying $x_1$ may move the candidate into the acceptance region (in blue).

resumes.

**Definition 4.1** (Mathematical formulation of Strategic LLM Manipulation). An *LLM manipulation* is a random function $L : \mathbb{R}^d \to \mathbb{R}^d$ characterized by a series of (not necessarily independent) real-valued random variables $\chi_1, \chi_2, \ldots, \chi_{d_1}$. When called upon a feature vector $\boldsymbol{x} = [x_1, \ldots, x_{d_1}, c_1, \ldots, c_{d_2}]$,

1. $L$ *replaces* each $x_i$ with a value drawn from $\chi_i$ for $1 \leq i \leq d_1$.

2. $L$ *preserves* the value of $c_j$ for $1 \leq j \leq d_2$.

That is,

$$L([x_1, \ldots, x_{d_1}, c_1, \ldots, c_{d_2}]) = [\chi_1, \ldots, \chi_{d_1}, c_1, \ldots, c_{d_2}].$$

Our formulation of LLM manipulations is based on our observations that LLMs can standardize style features such as writing quality, vocabulary, and organization — to the point that the original values are irrelevant and are redistributed according to a distribution dependently only on the LLM. On the other hand, candidates would like LLMs to preserve their fundamental features. Changes to their fundamental features may be extremely costly to the candidate, as hirers may decide to blacklist or dismiss dishonest candidates. The prompts for our experiments also work to elicit this outcome: to minimize the chances of hallucination, our prompt states explicitly that is "imperative that the new resume do not add any facts that are not in the original resume". We manually inspected some sampled outputs to confirm that generated outputs were free of hallucinations, though not all outputs were inspected. This perspective is informed by our empirical observations detailed in Section 3. Our experiments first indicated that LLM manipulations effectively overwrite writing style attributes. This is captured in our model, where the style features are redistributed according

to a fixed random distribution. We remark that $L$ could represent a single use or multiple uses of an LLM to improve a resume.

### 4.1.1. HIRING SCHEMES

In our hiring scheme, we define the *Hirer* who is making the hiring decisions and the *Candidate* who is applying for the job. Our work focuses on job positions receiving large volumes of applications: for this reason, we assume that the Hirer screens each candidate's resume by assigning each a real-valued score. More specifically, our model assumes that the Hirer uses some fixed scorer to evaluate the candidates resumes (in our experiments, we use Resume Matcher as the scorer). We represent this scorer as a function $s : \mathbb{R}^d \to \mathbb{R}$. We make no assumptions about $s$ other than that it is monotonically non-decreasing.

Note that we assume that the Hirer has no additional control over $s$. In practice, employers have little control over the scorers purchased from applicant screening software providers at the candidate screening stage. They can, at best, choose the best (e.g., most accurate) scorer available and tweak it accordingly. Under our model, as the Hirer does not have the resources to manually filter resumes, the resume score is how the Hirer decides which candidates move on to the next stage of hiring. This reflects the widespread usage of applicant tracking systems by employers.

We assume that each Candidate group $g$ has access to its own LLM, $L_g$. Likewise, the Hirer also has access to their own LLMs that are separate from the Candidates'.

### 4.2. Traditional Hiring with LLM Manipulations

We now introduce our strategic LLM classification game for traditional hiring with Candidate LLM manipulation. A candidate can use the LLM available to their group ($L_g$) to manipulate their resume — the candidate then chooses which of these two resumes (original or manipulated) to submit to the Hirer.

The Hirer determines a threshold $\tau \in \mathbb{R}$ and accepts candidates with scores equal to or greater than the threshold. Namely, the Hirer decision regarding a candidate with a submitted resume $\boldsymbol{x}'$ is $f_\tau(\boldsymbol{x}') = \mathbb{1}[s(\boldsymbol{x}') \geq \tau]$. We assume that there are many candidates, in which case minimizing false positives is a natural hiring objective, as the cost of interviewing or hiring unqualified candidates outweighs the cost of overlooking qualified ones. False positive has been studied in the context of fairness (e.g., Cohen et al. (2020), Blum et al. (2022)), and strategic classification (e.g., Ahmadi et al. (2022), Shao et al. (2023)). In this vein, we introduce the No False Positives Objective.

**Definition 4.2** (No False Positives). The *No False Positives Objective* is achieved when the Hirer maximizes true

positive rate (TPR) subject to no false positives. The optimization problem is:

$$\begin{array}{ll} \text{maxmize}_\tau & \text{TPR}(\tau) \\ \text{subject to} & \text{FPR}(\tau) = 0 \end{array}$$

where

$$\text{TPR}(\tau) = \mathbb{P}(f_\tau(\boldsymbol{X}') = 1 \mid Y = 1) \quad \text{and}$$
$$\text{FPR}(\tau) = \mathbb{P}(f_\tau(\boldsymbol{X}') = 1 \mid Y = 0).$$

We let $\tau^*$ denote the minimum threshold in the solution set.

Our work specifically focuses on classifiers that optimize true positive rates: this approach will specifically inform our further study of disparities between groups in Section 5. We aim to satisfy a specific case of *Equalized Odds* (Hardt et al., 2016b) when the false positive rate is fixed at zero, which is a special case of equalized false positive rates across groups.

---

**Definition 4.3** (TRADITIONAL Hiring Scheme under LLM Manipulation). The Hirer and the Candidate play the following Stackelberg game.

1. The Hirer commits to a scorer $s$ and a threshold $\tau \in \mathbb{R}$, both unknown to candidates.

2. Each candidate $(\boldsymbol{x}, g, y)$ chooses to submit either their original resume $\boldsymbol{x}' = \boldsymbol{x}$ or their LLM-manipulated resume $\boldsymbol{x}' = L_g(\boldsymbol{x})$.

3. The Hirer accepts candidates according to the threshold classifier $f_\tau(\boldsymbol{x}') = \mathbb{1}\left[s(\boldsymbol{x}') \geq \tau\right]$.

Each player has the following payoffs:

1. The Candidate payoff is whether they are accepted: $\mathbb{1}\left[f_\tau(\boldsymbol{x}') = 1\right]$.

2. The Hirer's payoff is defined according to the No False Positives Objective (Definition 4.2).

---

**Remark.** Unlike classic strategic classification, our game does not *directly* assume that the Candidate has perfect knowledge about $f_\tau$, as hiring schemes are often opaque. However, we assume that candidates know which of the two versions (unmanipulated and manipulated) of their resume will score higher. Additionally, since writing a prompt in an LLM is very easy, we assume it has negligible cost and that each candidate will use the more advanced LLM if they have access to it (i.e., a candidate from the privileged group will not use $L_U$). A best-responding candidate in group $g$ will therefore submit

$$\boldsymbol{x}'_g = \operatorname*{argmax}_{\boldsymbol{z} \in \{\boldsymbol{x}, L_g(\boldsymbol{x})\}} s(z).$$

In our model, candidates do not incur costs for prompting their LLM for the manipulation or for selecting the better

application. This is in contrast to prior work in strategic classification, where manipulations, such as getting multiple credit cards, require time and effort (Hardt et al., 2016a). Our model does however separate $L_P$ and $L_U$: this is equivalent to how privileged groups in prior works are given larger budgets when costs are incurred (Milli et al., 2019).

The Hirer does not know and may not infer whether a resume has been manipulated or from which group a resume comes. Rather, the Hirer must use the same scoring scheme and threshold for all candidates.

## 5. Disparities in Traditional Hiring with Unequal Candidate LLM Manipulation

In this section, we show that, under a traditional hiring scheme, disparities in LLM qualities between candidate groups can lead to disparities in hiring outcomes. We begin by defining a useful metric for disparity in hiring outcomes. Since we assume that groups $P$ and $U$ have the same unmanipulated feature vector distribution, we define the resume outcome disparity as follows.

**Definition 5.1.** Given a resume feature vector $\boldsymbol{x} \in \mathbb{R}^d$, the *resume outcome disparity* $\Delta$ is defined as

$$\Delta(\boldsymbol{x}) = \mathbb{P}_{L_P}\left(f_\tau(\boldsymbol{x}'_P) = 1\right) - \mathbb{P}_{L_U}\left(f_\tau(\boldsymbol{x}'_U) = 1\right),$$

where $\boldsymbol{x}'_g = \operatorname{argmax}_{\boldsymbol{z} \in \{\boldsymbol{x}, L_g(\boldsymbol{x})\}} s(\boldsymbol{z})$ for $g \in \{P, U\}$.

Observe that if the original unmanipulated resume is already accepted (that is, $f_\tau(\boldsymbol{x}) = 1$), then $\Delta(\boldsymbol{x}) = 0$.

To capture the differences in the output quality of different LLMs, we use the notion of multivariate stochastic dominance (Levhari et al., 1975).

**Definition 5.2** ((Levhari et al., 1975)). Let $Z_1, Z_2$ be random variables over $\mathbb{R}^d$. For any $a \in \mathbb{R}^d$, let $F_k(a) = \mathbb{P}(Z_k \leq a)$, where $\leq$ denotes component-wise order. We say that $Z_1$ *stochastically dominates* $Z_2$ if for any open lower set $S \subseteq \mathbb{R}^d$,

$$\int_S dF_1 \leq \int_S dF_2.$$

This is a generalization of (first-order) univariate stochastic dominance to multivariate distributions. Intuitively, stochastic dominance requires that the generalized CDF of $Z_1$ must always be "less" than the generalized CDF of $Z_2$. Stochastic dominance induces a partial order over multivariate random variables. Furthermore, we use the following key property about stochastic dominance.

**Lemma 5.1** ((Levhari et al., 1975)). $Z_1$ stochastically dominates $Z_2$ if and only if for every non-decreasing function $u$,

$$\mathbb{E}[u(Z_1)] \geq \mathbb{E}[u(Z_2)].$$

We use this definition to define our ordering over LLM quality.

**Definition 5.3.** Let $L_1, L_2$ be LLM manipulations. We say that $L_1$ *dominates* $L_2$ ($L_1 \succeq L_2$) if for all $\boldsymbol{x} \in \mathbb{R}^d$, $L_1(\boldsymbol{x})$ stochastically dominates $L_2(\boldsymbol{x})$.

Informally, an LLM $L_1$ may be considered "better" than $L_2$ if it stochastically dominates $L_2$ on each input, indicating that $L_1$ has a greater likelihood of feature improvement than $L_2$. Note that this only implies that $L_1$ *tends* to produce a better output than $L_2$; $L_2$ may produce a better output than $L_1$ on certain realizations of their stochastic outputs.

**Remark.** To simulate the absence of access to an LLM in our strategic classification game, it is helpful to artificially define a "null LLM" ($L_\varnothing$) that is dominated by all other LLMs. We might informally conceptualize $L_\varnothing$ as having random variables $\chi_i = -\infty$ for $1 \leq i \leq d_1$. This will allow us later to conceptualize traditional hiring as a special case of our proposed TWO-TICKET hiring scheme in which the Hirer deploys the null LLM.

We now show that, under this definition, using a better LLM on the same resume leads to a better hiring outcome.

**Theorem 1.** Suppose $L_P \succeq L_U$. Then for all $\boldsymbol{x} \in \mathbb{R}^d, \Delta(\boldsymbol{x}) \geq 0$.

*(Proof Sketch).* Since $\mathbb{P}_{L_g}(f_\tau(\boldsymbol{x}'_g) = 1) = \mathbb{E}_{L_g}[f_\tau(\boldsymbol{x}'_g)]$ and $f_\tau$ is non-decreasing, we can apply Lemma 5.1 to show that $\mathbb{P}_{L_P}(f_\tau(\boldsymbol{x}'_P) = 1) \geq \mathbb{P}_{L_U}(f_\tau(\boldsymbol{x}'_U) = 1)$. $\square$

This disparity in resume outcomes naturally leads to disparity in group outcomes. Under the No False Positives Objective, it is natural to measure group outcomes by comparing groups' true positive rates. We denote the TPR over a group $g$ as

$$\text{TPR}_g = \mathbb{P}\left(f_\tau(\boldsymbol{X}'_g) = 1 \mid Y = 1, G = g\right).$$

To address fairness, we define the disparity between the TPRs of two groups. This fairness notion has been studied previously in the context of strategic classification (e.g., (Keswani & Celis, 2023)).

**Definition 5.4.** The *TPR disparity* $\Delta_{\text{TPR}}$ is defined as

$$\Delta_{\text{TPR}} = \text{TPR}_P - \text{TPR}_U.$$

Having defined the TPR disparity, we show that qualified candidates from the privileged group have a higher (or equal) probability of being accepted compared to qualified candidates from the unprivileged group.

**Corollary 1.** Suppose $L_P \succeq L_U$. Then, $\Delta_{\text{TPR}} \geq 0$.

*(Proof Sketch).* This follows from applying Theorem 1 over candidates with $Y = 1$. $\square$

## 6. Combating LLM Disparities: Two-Ticket Scheme

To counteract the disparity in hiring outcomes due to unequal LLM access, we propose a modified hiring scheme where the Hirer performs their own round of LLM manipulation over the possibly manipulated applications. Our motivating experiments (Section 3) show that running a resume through a high-quality LLM twice changes the resume much less on the second run than on the first. Consequently, we show that bestowing both groups with the benefit of a round of high-quality LLM manipulation can help level the playing field.

### 6.1. Two-Ticket Scheme

We present the modified strategic classification game under the TWO-TICKET scheme. This scheme is identical to TRADITIONAL hiring except that the Hirer now uses their own LLM ($L_H$) to manipulate each submitted resume. The Hirer then scores the best of these two versions to determine whether to accept each candidate.

---

**Definition 6.1** (TWO-TICKET Hiring Scheme under LLM Manipulation)**.**

1. The Hirer commits to a scorer $s$ and a threshold $\tau \in \mathbb{R}$, both unknown to candidates and some LLM $L_H$.[a]

2. Each candidate $(\boldsymbol{x}, g, y)$ chooses to submit either their original resume $\boldsymbol{x}' = \boldsymbol{x}$ or their LLM manipulated resume $\boldsymbol{x}' = L_g(\boldsymbol{x})$.

3. The Hirer chooses to consider the higher scoring resume among the submitted resume $\boldsymbol{x}'' = \boldsymbol{x}'$ and the LLM-manipulated submission $\boldsymbol{x}'' = L_H(\boldsymbol{x}')$.

4. The Hirer accepts candidates according to the threshold classifier $f_\tau(\boldsymbol{x}'') = \mathbb{1}\left[s(\boldsymbol{x}'') \geq \tau\right]$.

Each player then has the following payoffs:

1. The candidate payoff is the probability that they are accepted: $\mathbb{P}_{L_H}\left(f_\tau(\boldsymbol{x}'') = 1\right)$.

2. The Hirer's payoff is defined according to the No False Positives Objective (Definition 4.2).

---
[a]Text in blue distinguishes our TWO-TICKET hiring scheme from the TRADITIONAL hiring scheme.

---

In practice, the Hirer scores both the submitted resume and the Hirer LLM-manipulated resume, and accepts the candidate if one of the scores passes the threshold. This is where our name "Two-Ticket Hiring" comes from: each candidate

is essentially given two avenues to acceptance.[6] This is equivalent to the above definition: our chosen presentation emphasizes the symmetry of the Hirer's LLM manipulation and the Candidate's LLM manipulation.

While LLM manipulations generally improve resume quality, there is a chance that they can decrease a candidate's score (Section 3). To ensure that candidates are not unfairly harmed by LLM manipulations, we safeguard against this possibility by requiring the Hirer to evaluate the maximum of the candidate's submitted and its Hirer-manipulated version of each resume.

The TRADITIONAL hiring game (Definition 4.3) can be considered a special case of the TWO-TICKET hiring game (Definition 6.1), where $L_H$ is the null LLM discussed in Section 5; in which case $L_H(x')$ is always inferior to, and discarded in favor of, the original submitted resume $x'$. Thus, we compare the behavior of two different TWO-TICKET games: the game under TRADITIONAL hiring (Definition 4.3) and the game under TWO-TICKET hiring schemes (Definition 6.1), which now differ only in the Hirer's choice of LLM and threshold in our formalization.

## 6.2. Guaranteed Two-Ticket Improvements

We now prove that under natural conditions, a TWO-TICKET hiring scheme can decrease the resume outcome disparity between the two groups, leading to improvement in accuracy and fairness.

For $k \in \{1, 2\}$, we define Hiring Scheme $k$ to be the TWO-TICKET scheme using Hirer LLM $L_H^{(k)}$ and scheme-dependent threshold $\tau^{(k)}$, resulting in deployed classifier $f^{(k)}$. We let $\Delta^{(k)}(x)$ and $\Delta_{\text{TPR}}^{(k)}$ denote the resulting resume and group outcome disparity respectively for Hiring Scheme $k$. To compare the TRADITIONAL hiring scheme with the TWO-TICKET hiring scheme, we denote the TRADITIONAL hiring scheme as $k = 1$ with $L_H^{(1)} = L_\varnothing$. With this definition, we are guaranteed that $L_H^{(2)} \succeq L_H^{(1)}$. Note that the following results still apply if Hiring Scheme 1 is a TWO-TICKET hiring scheme with a non-null Hirer LLM.

Our results apply when the same threshold can be used to achieve the No False Positives Objective across both schemes. We show that when the Hirer chooses LLM that are stochastically dominated by the privileged group's LLM, then this choice is sufficient (though not necessary) to ensure that the optimal threshold remains the same across hiring schemes.

**Lemma 6.1.** If $L_P \succeq L_H^{(1)}, L_H^{(2)}$, then $\tau^{*(1)} = \tau^{*(2)}$.

---

[6]Traditional hiring can be analogously thought of as "One-Ticket" hiring, as the candidate's submitted resume is their only avenue to acceptance.

Before giving our main results on outcome disparity, we reformulate the probability of acceptance under the TWO-TICKET hiring scheme.

**Lemma 6.2.** For Hirer LLM $L_H$ and threshold $\tau$, the probability that a candidate $(x, g, y)$ is accepted is

$$\mathbb{P}_{L_g, L_H}(f_\tau(x_g'') = 1) =$$
$$1 - \mathbb{1}\left[s(x) < \tau\right] \cdot \mathbb{P}_{L_g}\left(s(L_g(x)) < \tau\right) \cdot \mathbb{P}_{L_H}\left(s(L_H(x)) < \tau\right).$$

*(Proof Sketch).* This follows from Definition 6.1, using that $L_H(x_g')$ and $x_g'$ are conditionally independent given $x$. $\square$

Using Lemma 6.2, we derive our main result showing the improvement in resume outcome disparity by shifting from a TRADITIONAL to a TWO-TICKET scheme.

**Theorem 2.** Let $\tau^{*(1)} = \tau^{*(2)}$, $L_P \succeq L_U$, and $L_H^{(2)} \succeq L_H^{(1)}$. Then for all $x \in \mathbb{R}^d$, $\Delta^{(2)}(x) \leq \Delta^{(1)}(x)$.

**Remark.** Lemma 6.1 provides a simple and sufficient but not necessary condition that $\tau^{*(1)} = \tau^{*(2)}$ under the No False Positives Objective. In fact, Theorem 2 applies under *any* Hirer objective so long as the optimal deployed threshold is the same for Hiring Scheme 1 and 2.

Under the No False Positives Objective, the decrease in resume outcome disparity immediately implies a decrease in group outcome disparity and an increase in accuracy (or equivalently, true positive rate under the No False Positives Objective) for both groups.

**Corollary 2.** Let $\text{TPR}_g^{(k)}$ denote the true positive rate over group $g$ under Hiring Scheme $k$. Let $\tau^{*(1)} = \tau^{*(2)}$, $L_P \succeq L_U$, and $L_H^{(2)} \succeq L_H^{(1)}$. Then

1. $\left|\Delta_{\text{TPR}}^{(2)}\right| \leq \left|\Delta_{\text{TPR}}^{(1)}\right|$.

2. $\text{TPR}_g^{(2)} \geq \text{TPR}_g^{(1)}$ for $g \in \{P, U\}$.

3. $\text{TPR}^{(2)} \geq \text{TPR}^{(1)}$.

Since the threshold $\tau^*$ already prevents false positives (Definition 4.2), (3) also implies that accuracy does not decrease.

## 6.3. The $n$-Ticket Scheme and Group Dependence Bias Mitigation

While the TWO-TICKET scheme helps mitigate disparities, it may not be sufficient since the privileged group still has the advantage of a "better" first ticket, increaseing their acceptance probability. We therefore propose generalizing the idea to an $n$-*Ticket Hiring Scheme*. Let $L_H^n$ represent the application of the two-ticket scheme $n \in \mathbb{N}$ times using LLM $L_H$ (the $n$-ticket scheme). That is, for any $x \in \mathbb{R}^d$,

Step 3 in Definition 6.1 is repeated $n$ times, each time after the first with $\boldsymbol{x}''$ as the submitted resume. We will show that by applying the $n$-ticket scheme, the outcome becomes independent of group membership.

We start by defining a contraction operator and stating Banach's Fixed Point Theorem, which will be useful in the proof of the main theorem in this section.

**Definition 6.2.** Let $(Z, d)$ be a metric space. A function $T : Z \to Z$ is a contraction operator if for some $k \in [0, 1)$,

$$d(T(z), T(z')) \leq k \, d(z, z') \quad \text{for all } z, z' \in Z.$$

**Theorem 3** (Banach's Fixed Point (Banach, 1922))**.** Let $T : Z \to Z$ be a contraction operator. Then,

- The equation $T(z) = z$ has a unique solution $z^* \in Z$.

- For any $z_0 \in Z$, $\lim_{n \to \infty} T^n(z_0) = z^*$. Furthermore, $|T^n(z_0) - z^*| \leq \mathcal{O}(k^n)$, where $k$ is the contraction coefficient.

Before we state the main theorem, we note that the $n$-ticket scheme is defined in such a way that, once an applicant has at least one (possibly LLM-manipulated) resume that receives a score above the threshold $\tau$, then they are guaranteed to be accepted: this arises from the fact that the Hirer considers the *best-scoring* of the applicant's $n$ "tickets". In the main theorem, we show that for an infinite number of tickets, the hiring outcome becomes independent of group membership.

**Theorem 4.** Let $\tau$ be the threshold used by the Hirer in the $n$-ticket scheme. If $L_H \succeq L_P \succeq L_U$, applying the $n$-ticket scheme and taking the limit as $n \to \infty$, then any applicant $\boldsymbol{x} \in \mathcal{D}$ is guaranteed to receive a group-independent outcome, $o = o(\boldsymbol{x}, L_H) \in \{0, 1\}$. Furthermore, there exists $k_{\boldsymbol{x}} \in [0, 1)$, dependent on $L_H$ and $\boldsymbol{x}$, such that

$$|\, \mathbb{P}(f_\tau(L_H^n(L_g(\boldsymbol{x}))) = 1) - o| \leq \mathcal{O}(k_{\boldsymbol{x}}^n).$$

To prove the theorem, we show that providing an additional ticket for an applicant $\boldsymbol{x}$ is a contraction operator on $[0, 1]$, independent of group membership, and show the existence of a Banach fixed point (See Appendix F for full proof).

The theorem implies that by using the $n$-ticket scheme with an LLM as least as strong as the privileged group, the Hirer can significantly reduce any group-dependency bias in the hiring scheme, and the probability of not receiving the right outcome for the applicant drops exponentially in the number of tickets. As a corollary, the probability of a disparity in outcomes of candidates with the same feature vector but different groups and the TPR disparity drop exponentially with $n$.

**Corollary 3.** If $L_H \succeq L_P \succeq L_U$, then for every unmodified resume $\boldsymbol{x} \in \mathbb{R}^d$, there exists $k_{\boldsymbol{x}} \in [0, 1)$ that depends on $L_H$ and $\boldsymbol{x}$ such that for any $n \geq 2$, $\mathbb{P}(f_\tau(L_H^n(L_U(\boldsymbol{x}))) \neq f_\tau(L_H^n(L_P(\boldsymbol{x})))) \leq O(k_{\boldsymbol{x}}^n)$. Hence,

1. $|\Delta_{TPR}^{(n)}| \leq \mathcal{O}(k^n)$, where $k = \max_{\boldsymbol{x}} k_{\boldsymbol{x}}$.

2. $\text{TPR}_g^{(n)} \geq \text{TPR}_g^{(n-1)}$ for $g \in \{P, U\}$ and $n > 1$.

3. $\text{TPR}^{(n)} \geq \text{TPR}^{(n-1)}$ for every $n > 1$.

Since the threshold $\tau^*$ already prevents false positives (Definition 4.2), (3) also implies that accuracy does not decrease.

## 7. Empirical Validation: Resume Selection in the Technology Sector

In this section, we empirically validate our theoretical results by closely simulating a hiring scenario in which an employer has two positions to fill. We examined 520 resumes from the Djiini dataset (Drushchak & Romanyshyn, 2024), which includes resumes from the technology sector. Our sample consisted of equal parts UI/UX designers and project managers (PM). To replicate a real-world applicant tracking system, we used an open-source resume scorer, Resume Matcher,[7] to assign a relevance score (e.g., 0–100) for all resumes against a PM job description and a UX job description. We note that Resume-Matcher assigns its scores based on word-similarity metrics between the inputted resumes and target job descriptions.

We examine the GPT family of OpenAI models due to its widespread use. [8] We randomly assigned half of the resumes to group $P$ (privileged) and the remaining half to group $U$ (unprivileged). Only the candidates assigned to group $P$ could manipulate their resumes with the same model as the employer (GPT-4O). The resumes in group $U$ could only access GPT-3.5-TURBO, GPT-4O-MINI, or no LLM at all to edit the original resumes based on a resume improvement prompt. [9] Our theoretical results assume candidates to be best-responding, hence in our experiments, the candidates would submit the higher scoring resume between their choice of their LLM manipulated and original resume.

The Hirer learns a threshold that maximizes the true positive rate while minimizing the false positive rate (this objective

---

[7]This open-source resume scorer is designed to mimic applicant tracking systems that many hiring companies use for ranking applicant relevance (Purcell, 2024). Job applicants can use these ATS tools improve their resume relevance. To the best of our knowledge, ResumeMatcher is the only open-source ATS tool available (https://resumematcher.fyi/).

[8]ChatGPT is reported to have 2.4 billion monthly visits in March 2024, 10 times the next most popular platform (Zhu, 2024).

[9]Appendix D includes prompts, model versions, and costs.

| Condition | Method | PM Role | | UX Designer Role | |
|---|---|---|---|---|---|
| | | TPR | TPR Disparity | TPR | TPR Disparity |
| $U$: No LLMs, $P$:GPT-4O | Traditional | $0.11 \pm 0.004$ | $0.10 \pm 0.005$ | $0.22 \pm 0.008$ | $0.27 \pm 0.006$ |
| | Two-Ticket | $0.14 \pm 0.005$ | $0.05 \pm 0.005$ | $0.38 \pm 0.008$ | $0.01 \pm 0.007$ |
| $U$: GPT-3.5, $P$:GPT-4O | Traditional | $0.09 \pm 0.004$ | $0.09 \pm 0.005$ | $0.26 \pm 0.010$ | $0.15 \pm 0.008$ |
| | Two-Ticket | $0.11 \pm 0.004$ | $0.08 \pm 0.005$ | $0.30 \pm 0.010$ | $0.08 \pm 0.007$ |
| $U$: GPT-4O-MINI, $P$:GPT-4O | Traditional | $0.12 \pm 0.004$ | $0.04 \pm 0.005$ | $0.33 \pm 0.010$ | $0.00 \pm 0.007$ |
| | Two-Ticket | $0.13 \pm 0.007$ | $0.03 \pm 0.010$ | $0.36 \pm 0.010$ | $-0.01 \pm 0.008$ |

*Table 1.* Resume screening results where Groups $P$ and $U$ have access to various models of GPT family models for a PM and Design Job description respectively. Results are presented with 95% CIs computed over 500 train-test splits.

approximates the objective of no false positives in Definition 4.2). For the TRADITIONAL hiring scheme, the Hirer directly uses the "submitted" resumes from the two groups. For the TWO-TICKET scheme, the Hirer also manipulates each submitted resume with the employer model (GPT-4O). The Hirer then acts on the higher-scoring resume between the submitted and Hirer-manipulated versions of the resume. In both schemes, the Hirer has no knowledge about which individuals belong to which group; thus, membership-based fairness interventions cannot be applied to our setting.

Table 1 shows the empirical verification of our theoretical results, comparing the performance of the TRADITIONAL hiring scheme and the TWO-TICKET hiring scheme validated for both job descriptions over 520 resumes each. True positive rates were improved and TPR disparities were reduced by the TWO-TICKET scheme. The improvement was strongest when group $U$ used a weaker modification (i.e., GPT-3.5-TURBO or no LLM). When group $U$ used a similar level LLM (e.g., GPT-4O-MINI), there was no improvement in the TPR disparity.[10] Our results demonstrate that our proposed scheme helps better discern qualified candidates from candidates using stronger LLM manipulations [11].

## 8. Discussion

Our work is a first step toward understanding and designing better selection algorithms under stochastic LLM manipulations. Similarly to prior work (Hu et al., 2019a), we show that members of the privileged group are more easily admitted or hired. Here, privilege includes both access to more advanced LLMs and knowledge of the performance of different LLMs. While we focus on the setting of hiring, our TWO-TICKET scheme can be generalized to many other scenarios in which candidates can manipulate their materials with LLMs.

In our model, the Hirer does not know *a priori* whether a candidate has manipulated their resume. Our theoretical results imply that using the TWO-TICKET scheme, both the TPR and the TPR disparity are improved even without this knowledge. Specifically, our theoretical results suggest that this improvement is greatest when the Hirer deploys an LLM that is as strong as possible, while being weaker than the candidate's strongest LLM. Therefore, careful thought and evaluation must be used when applying our TWO-TICKET scheme in practice. Although our findings focus on manipulations that preserve the distinguishability between negative and positive labels within our candidate screening task, future work should investigate the full spectrum of choice of LLM. This is particularly impactful when companies may introduce increasingly premium LLM services.

We assume that companies use LLM APIs such that queries are not stored by the company, to preserve candidate's data privacy. We do not anticipate the resume manipulation itself to be any less private than the companies storing a candidate's resume. A differentially private threshold function can be used to ensure differential privacy (Bun et al., 2024).

As our work focused on simple prompts to capture a low-effort (zero-cost) manipulation, future work should address the variable behavior of LLMs that can arise from using different or more prescriptive prompts. Although we provide a theoretical guarantee for improvements in our TWO-TICKET scheme, relaxing the condition that the optimal threshold does not change could help establish stronger guarantees. Our experiments use the only open source ATS system available; future audits of actual hiring systems should test black-box and human-in-the-loop systems.

## Impact Statement

This paper is a theoretical work that examines the effect of an increasing number of job applications being produced by generative AI. Notably, we do not advocate for hiring systems using AI but study the problem of candidates using AI to modify their resumes. Rather, we developed this project

---

[10]At the time of our submission, GPT-4O-MINI is offered for free by OpenAI. However, before GPT-4O-MINI was released (after GPT-4O was released), GPT-3.5-TURBO was the free model offered. In the future, LLM providers may offer new versions of models where the paid version is much better than the free version.

[11]Our code and data is available at https://github.com/heyyjudes/llm-hiring-ecosystem

to address the potential downstream impacts of generative AI.

Nonetheless, the subsequent hiring decisions made by humans may introduce further disparities that may affect employee satisfaction in the long run. In future work, we plan to do an empirical study of the impact of LLM-aided application materials across a variety of industries in consultation with economists.

At the moment, we also do not foresee negative consequences to our analysis at this time. Our work by no means is comprehensive in studying the allocation of opportunities in the era of generative AI. We hope that future works continue to examine this area of sociotechnical AI safety.

## Acknowledgments

The authors thank Omer Reingold for his helpful discussion. The authors are supported by the Simons Foundation Collaboration on the Theory of Algorithmic Fairness, the Sloan Foundation Grant 2020-13941, and the Simons Foundation investigators award 689988. Connie and Jack are grateful participants of the CURIS program that facilitated matching undergraduates to this project.

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

| Condition | Method | PM Role Accuracy | UX Designer Role Accuracy |
|---|---|---|---|
| $U$:No LLMs, $P$:GPT-4O | Traditional | $0.548 \pm 0.003$ | $0.633 \pm 0.005$ |
| | Two-Ticket | $0.563 \pm 0.004$ | $0.689 \pm 0.005$ |
| $U$:GPT-3.5, $P$:GPT-4O | Traditional | $0.543 \pm 0.003$ | $0.629 \pm 0.006$ |
| | Two-Ticket | $0.551 \pm 0.003$ | $0.650 \pm 0.006$ |
| $U$:GPT-4O-MINI, $P$:GPT-4O | Traditional | $0.554 \pm 0.003$ | $0.662 \pm 0.005$ |
| | Two-Ticket | $0.561 \pm 0.003$ | $0.677 \pm 0.005$ |

*Table 2.* Accuracy results for experiments conducted in Table 1. We see improvements in accuracy at 95% confidence.

## A. Full Related Work

**Fairness in Algorithmic Hiring** Audits of hiring systems have consistently found discrimination in outcomes based on race, gender, and age (Bertrand & Mullainathan, 2004; Kline et al., 2022). Raghavan et al. (2020) study the screening stage of the hiring algorithms and connect legal perspectives with algorithmic approaches to mitigate the disparate impact. Mitigating biases has also been described as crucial to the "ethicality of the AI tool design" (Hunkenschroer & Luetge, 2022). In terms of proposed solutions, (Lin et al., 2021) suggest "augmentation-based" interventions where AI-assisted decisions can best achieve equitable outcomes. A key assumption of prior work is access to (explicit or inferred) group membership. In our work, the hiring side has no knowledge of the group membership of candidates, yet we can mitigate bias nevertheless.

**Strategic Classification** Hardt et al. (2016a) introduce strategic classification as a Stackelberg game to address the impact of manipulative tactics on classification problems. We draw on several later works that provided a modified strategic classification game that models disparities in manipulation abilities (Hu et al., 2019b; Milli et al., 2019; Chen et al., 2020; Diana et al., 2024). Furthermore, we use techniques from Braverman & Garg (2020) to describe "random" classifiers in light of stochastic strategic manipulations. Similarly to several previous works (Ghalme et al., 2021; Cohen et al., 2024), we assume that the deployed classifier is unknown to the candidates. Finally, previous work of (Somerstep et al., 2024) has explored the use of casual strategic classification to also understand labor market dynamics: our work extends this by highlighting the effects of strategic behavior on employer and labor force outcomes.

**Behavior and Risks of Generative Models** Guidance counselors and career coaches alike now recommend using generative AI tools to help with application materials (Verma & Renjarla, 2024; Chamorro-Premuzic, 2024). However, recent research has highlighted significant risks. For example, LLMs have been shown to hallucinate, which may mislead employers (Huang et al., 2023), or memorize text, which can result in unintended plagiarism (Carlini et al., 2023). As unintended plagiarism is difficult for job applicants to detect using these tools, the benefits of applying LLMs to application materials may be stochastic.

## B. Additional Results

Figure 3 illustrates that using a simple job-agnostic prompt with an input resume significantly improves the scores computed by a resume screening system. Qualitatively, we observed a drastic improvement in writing quality (examples available in Appendix C.4); the LLMs were able to transform resumes mostly containing bullet points about the candidate's interests or skills into more effective, reworded resumes delineating prior roles. However, scores did not improve monotonically across resumes; some resume scores decreased after applying LLM manipulation (Figure 4a).

### B.1. Accuracy

In addition to observed improvements in TPR and TPR Disparity values from our experiments in Section 6.3, our experiments also yielded improvements in accuracy (Table 2).

### B.2. Thresholds

The main results of Section 7 highlight the validity of our theoretical findings: namely, we show that a TWO-TICKET scheme can both improve a classifier's true positive rate (TPR) and reduce disparities in TPRs between privileged and

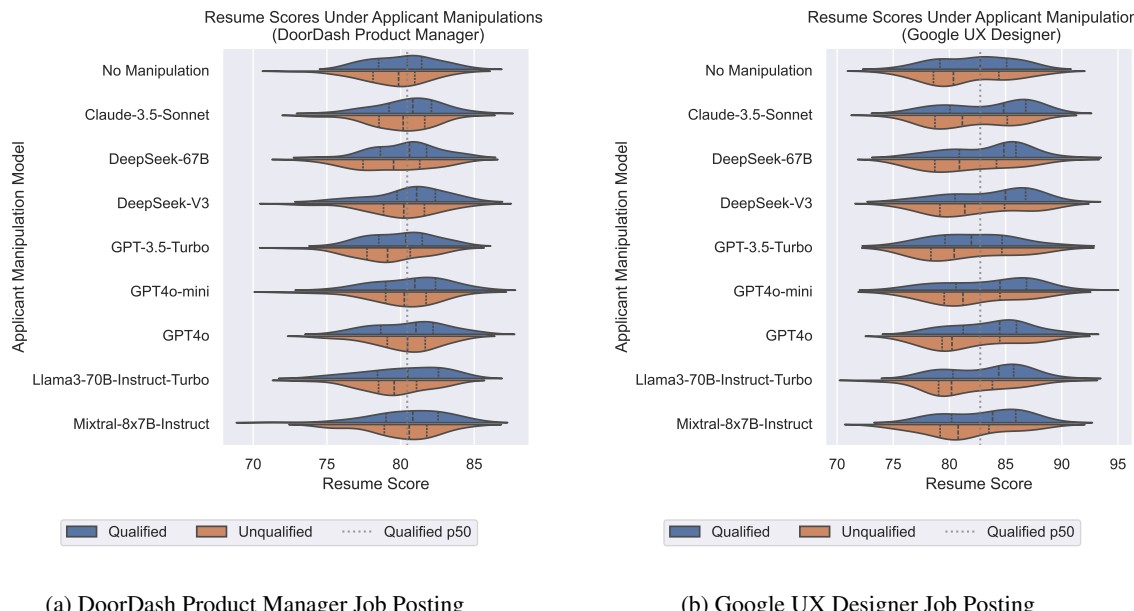

(a) DoorDash Product Manager Job Posting    (b) Google UX Designer Job Posting

*Figure 3.* Resume score distribution of 50 qualified (matching occupation) and 50 unqualified (different occupation) resumes before and after LLM manipulations for two job descriptions. Models such as GPT-4O and CLAUDE-3.5-SONNET and DEEPSEEK-V3 generate a distribution of unqualified resumes that is indistinguishable from qualified resumes without manipulation for the Product Manager job and significantly enhance the scores of the qualified resumes for the UX Designer position.

unprivileged groups when group membership is unknown to the classifier. In this section, we present further empirical findings to support our modeling assumptions. In particular, we validate the constancy of our No False Positives Objective thresholds, in accordance with Lemma 6.1. Since attempting to set the false positive rate (FPR) exactly to zero occasionally led to an empirically undefined TPR, we chose the threshold corresponding to the smallest FPR greater than zero to produce our main results in Section 7. While this approach does not strictly adhere to our No False Positives Objective, the resulting FPRs on the training set are still sufficiently close to zero, as seen in Table 3. Table 3 presents the FPRs incurred by the thresholds chosen for our experiments in Section 7. Overall, these results confirm that our experimental approximation of the No False Positives Objective is sufficiently similar.

| Method | False Positive Rate at 95% CI |
|---|---|
| Traditional | $0.0040 \pm 0.000260$ |
| Two-Ticket | $0.0014 \pm 0.000232$ |

*Table 3.* False Positive Rates on Training Sets for PM Role. Results are presented with 95% CIs computed over 500 train-test splits.

Tables 4 and 5 summarize our experimental results comparing the thresholds of our TRADITIONAL and TWO-TICKET schemes when both the Hirer and the user employ the same LLM, GPT-4O-MINI (i.e., $L_U = L_H$). While there is a slight, non-significant difference in thresholds between the one-ticket and two-ticket schemes in our main results (Table 2), introducing more randomness — by varying both the train-test splits and the privileged-unprivileged group assignments in each trial — led to more similar thresholds between the two schemes. This brought the results closer to satisfying the criteria outlined in Lemma 6.1. In contrast, the experiments in Section 7 used fixed privileged-unprivileged group assignments, and thresholds were measured across a range of fixed train-test splits.

We hypothesize that the slight difference between our assumptions from Lemma 6.1 and our empirical findings arise as a result of the random nature of the privileged-unprivileged group assignments and slight modification of the No False Positives Objective. Specifically, under our approximation of the No False Positives Objective, we find that our output threshold is determined by the top scoring UI/UX resumes (irrelevant of whether they have been manipulated or not). The fixed assignment of these "top" scoring UI/UX resumes to a "non"-manipulating group would hence result in a different threshold across the TRADITIONAL and TWO-TICKET scheme.

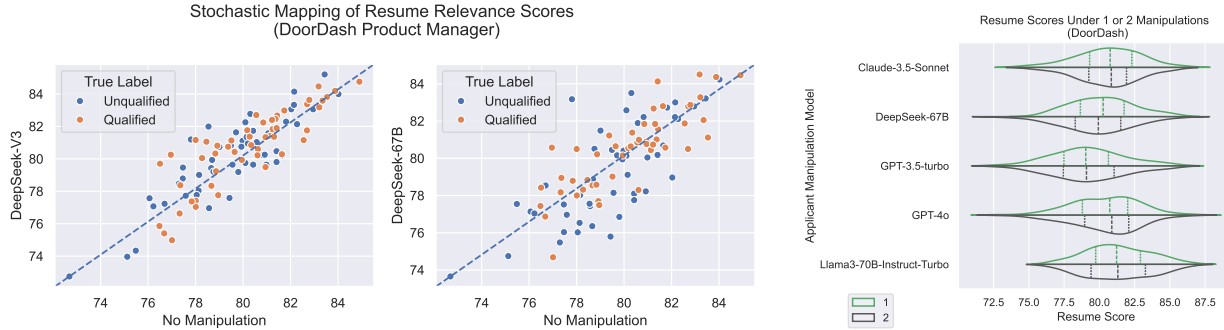

(a) Resume Score Improvement after applying LLM manipulation with different models.

(b) Resume Scores over Sequential LLM Manipulations

*Figure 4.* (a) Applying LLM manipulations to resumes result in stochastic outcomes: even when the average score increases (e.g., DEEPSEEK-V3), some resumes receive lower scores after manipulation. (b) Repeatedly using LLMs to enhance a resume results in stagnating improvements.

| Method | Threshold at 95% CI |
|---|---|
| Traditional | $84.3 \pm 0.040$ |
| Two-Ticket | $86.1 \pm 0.077$ |
| Difference | $1.78 \pm 0.086$ |

*Table 4.* Threshold Values for PM Role for Fixed Group and Train-Test Assignments

| Method | Threshold at 95% CI |
|---|---|
| Traditional | $85.3 \pm 0.05$ |
| Two-Ticket | $85.8 \pm 0.33$ |
| Difference | $0.46 \pm 0.06$ |

*Table 5.* Threshold Score Values for PM Role for Randomized Group and Train-Test Assignments

We motivate this with a simple example. Consider a scenario with a training set of four applicants applying for a PM Role. In this example, two of the applicants are UI/UX applicants who have been assigned to the unprivileged group, whereas the remaining PM applicants have been assigned to the privileged (who will manipulate their resume) group. Suppose our Hirer receives the following distribution of resume scores and their prior role, which it must optimize over under the TRADITIONAL scheme:

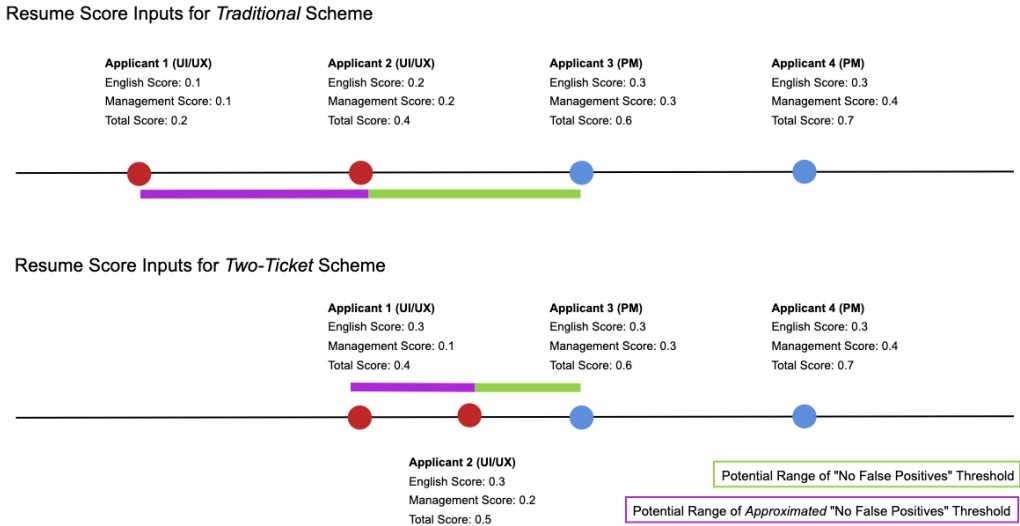

*Figure 5.* Example of Privileged-Unprivileged Assignment in the Training Set Resulting in Differing Thresholds Between the TRADI-TIONAL and TWO-TICKET Schemes.

Now, let us contrast this with the Hirer-manipulated resume scores over which the TWO-TICKET scheme optimizes. In this scenario, we assume that all English skills are mapped to a random variable with an expected value of 0.3 and zero standard deviation (for simplicity). Under the TWO-TICKET scheme, the scores of the UI/UX applicants "improve" as they have not previously modified their resumes: however, the scores of the PMs remain constant, as they have already modified their resumes. As shown, the resume scores that the Hirer optimizes over differ between the TRADITIONAL and TWO-TICKET schemes. This leads to different ranges of possible "No False Positive" thresholds for each scheme. On the contrary, when strictly adhering to a "No False Positive" objective, we observe that the change in the threshold is smaller. To further support this hypothesis empirically, Table 6 displays the maximum negative resume scores which the classifier optimizes over, regardless of whether the resumes were manipulated. As the scores fluctuate across trials and do not have a zero standard deviation, it is clear that these values are changing. In summary, the discrepancy between our "Constant Threshold" and the empirical results can be attributed to small differences between our theoretical constraints and actual methods—specifically, in approximating the "No False Positives" objective. While our analysis helps explain some of these discrepancies, we note that the magnitude of the differences is small enough that it does not significantly affect the experimental conclusions drawn in Table 1.

| Method | Maximum UI/UX Resume Score |
|---|---|
| Two-Ticket Scheme | $86.7 \pm 3.6 \times 10^{-14}$ |
| Traditional Scheme | $85.90 \pm 0.07$ |
| Difference | $0.75 \pm 0.07$ |

*Table 6.* Maximum UI/UX Resume Scores for PM Role

## C. Empirical Motivation: Additional Results

### C.1. Additional Jobs

We include scores across different models for additional job descriptions. In Figure 6 shows three additional PM jobs and how LLM manipulations to resumes affect the job relevance scores. For PM jobs, some job descriptions generated similar median values for both the qualified and unqualified groups (e.g., Apple Product Manager and Yelp Product Manager). For these jobs, we observed significant boosts in relevance scores for both groups by newer, premium language models.

We also include three additional job descriptions for UX Designers positions (Figure 7). For UX Designers, there's a specific set of skills required that always separates the qualified (UX) from the unqualified (PM) resumes. However, the improvement

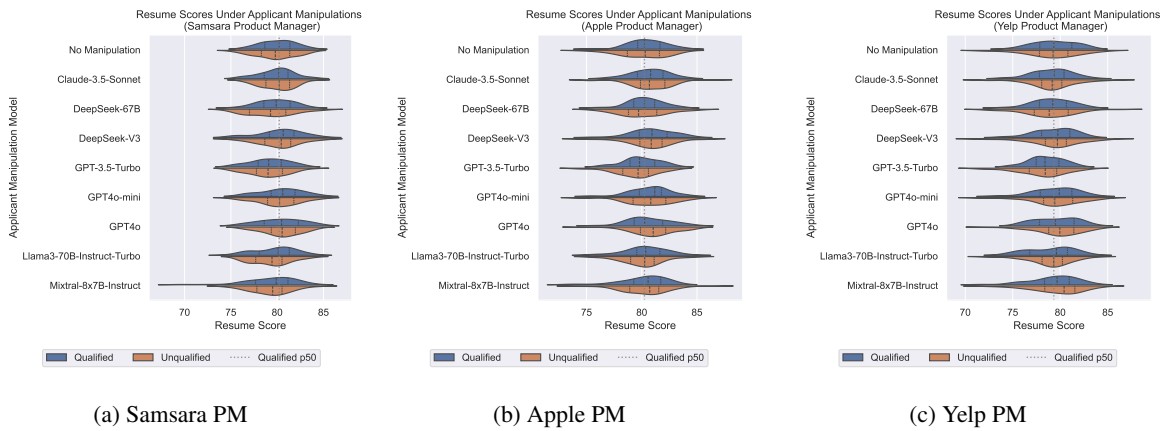

(a) Samsara PM          (b) Apple PM          (c) Yelp PM

*Figure 6.* Resume score distribution of 50 qualified (matching occupation) and 50 unqualified (different occupation) resumes before and after LLM manipulations for more Product Manager Job Descriptions

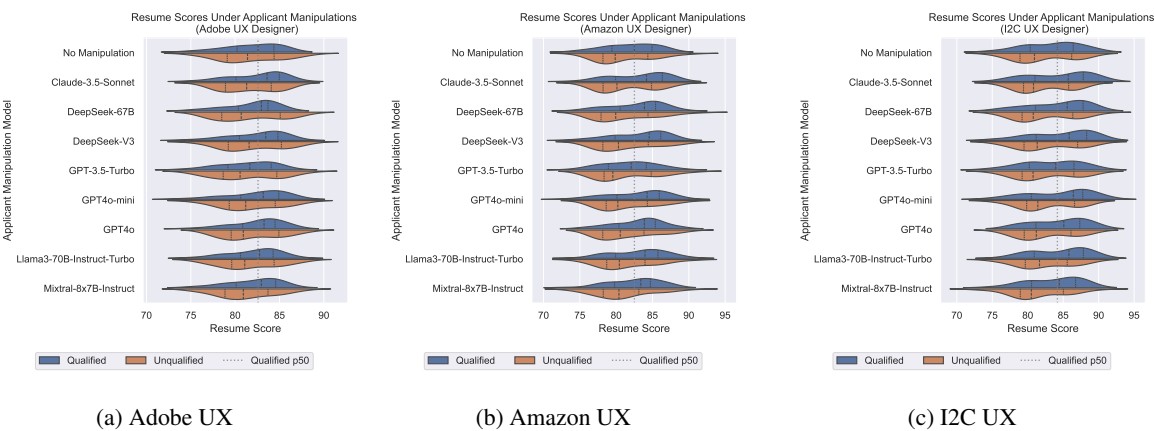

(a) Adobe UX          (b) Amazon UX          (c) I2C UX

*Figure 7.* Resume score distribution of 50 qualified (matching occupation) and 50 unqualified (different occupation) resumes before and after LLM manipulations for more UX Designer Job Descriptions

in scores is particularly stark for the qualified group which would introduce additional disparities.

## C.2. Homogenization: Model and Resume Similarity

While not directly applicable to our theoretical model, many recent works have studied the homogenization of language model outputs. Here we example how similar resumes are before and after LLM manipulation. We observe that experimentally that almost all models increase the similarities of resumes. This suggests that as more applicants turn to LLM tools, their resumes are growing more homogeneous to the employer.

We also examine similarity with the application of the relevance score system; do the same models find the same candidates good? To measure this, we find the correlation between 100 resume scores for each job description and plot a heat map between models (Figure 9). We observe a larger correlation between models for UX designer positions than for PM positions. This is likely because there are specific skills unique to UX designers that the unqualified resumes (PM resumes) do not have. We also see models from the same family (e.g. DEEPSEEK-67B and DEEPSEEK-V3, GPT4O and GPT4O-MINI) with higher correlation. This suggests models from the same family may modify resumes in a similar way.

## C.3. LLM Modifications

A key motivation for our work comes from our observations that LLMs can improve the written of quality of resumes, though at varying levels. In addition to Section 3, we here examine the exact disparities and rates of resume improvement.

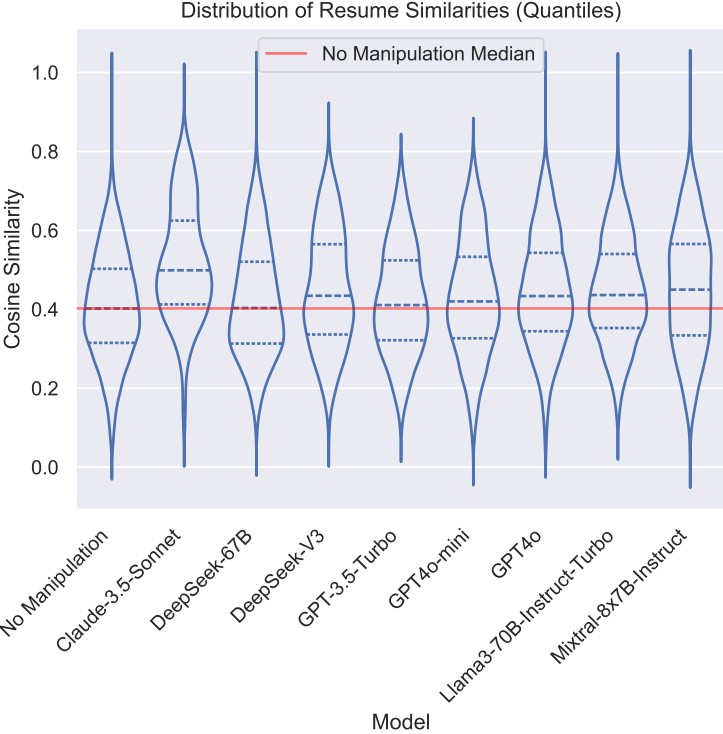

*Figure 8.* Distribution of cosine similarities (Sentence Embeddings ALL-MINILM-L6-V2) across all pairs of resumes. Compare to no manipulation, nearly all models increase the similarity of resumes, especially CLAUDE-3.5-SONNET.

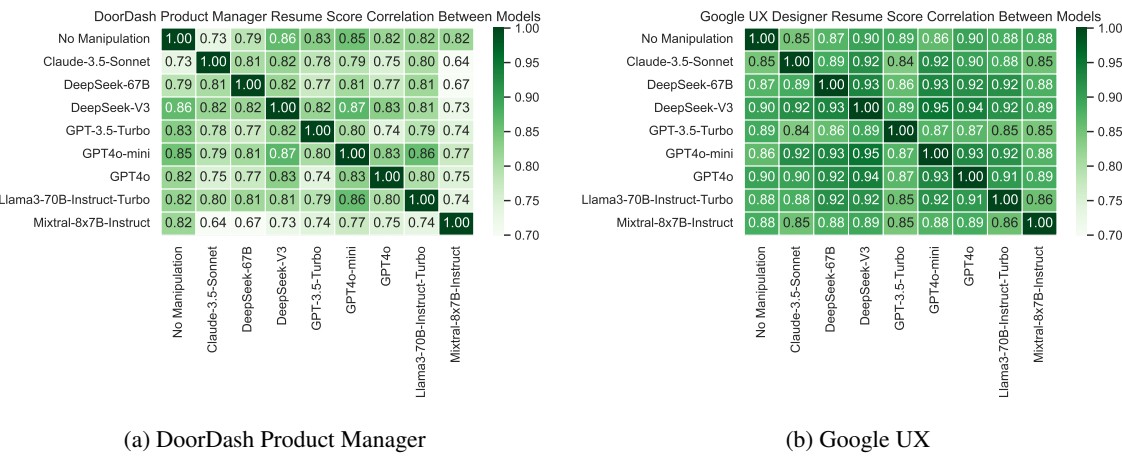

(a) DoorDash Product Manager          (b) Google UX

*Figure 9.* Pearson correlation between model scores of resumes. UX designer job descriptions had higher agreement between models.

We found that more resumes were improved by CHATGPT-4O as opposed to MIXTRAL-8X7B. 75% of our 520 tested resumes experienced increases in resume scores due to CHATGPT-4O modifications, while only half of the 520 resumes experienced increases in resume scores when modified by MIXTRAL-8X7B.

### C.4. Qualitative Analysis: Best-Responding Candidates

As noted in the main body of our paper, our theoretical and experimental findings assume that job candidates are "best-responding" and can choose to submit the higher scoring resume between their submitted and modified LLM version of their resume. In reality, however, we note that candidates don't have access to the score system deployed by a Hirer firm. Nevertheless, it is relatively easy, if not fast, to distinguish between the lower and higher-scoring resumes by hand. For example, consider the following snippets from two resumes - the first resume is an unmodified resume, with a score of 79.434, and the second is a modified resume, with an improved score of 81.882.

The "professional summary" section of one unmodified resumes is

> *Have skills in creating a prototype and choosing the methodology for leading the project. I have experience in creating BPMN diagrams. Also, I have worked with different PM tools and can highlight ClickUp as my favorite one.*

On the contrary, the "professional summary" section of the corresponding manipulated resumes is

> *Skilled Project Manager with expertise in document management for various project stages, creating prototypes, and selecting appropriate project methodologies. Hands-on experience with BPMN diagrams and adept in utilizing different project management tools, with a preference for ClickUp.*

The manipulated resume clearly exhibits more complete sentences and sophisticated vocabulary, indicating a perhaps better-scoring resume. For reference, the complete resumes are shown below.

### *Unmodified Resume (PM Score: 79.434)*

Have skills in maintaining documents for the stages of the project: Brief, Assumptions, WBS (WBS scheme, TTM, Calendar plan), Project Charter, MVP, Requirement Matrix, Risk Register, SRS, RACI, and Status Report.

Have skills in creating a prototype and choosing the methodology for leading the project. I have experience in creating BPMN diagrams. Also, I have worked with different PM tools and can highlight ClickUp as my favorite one.

**Coordinator of Club of Tolerance**
The Congress of National Communities of Ukraine — 2017—2019
I held weekly meetings with teenagers, where we discussed various topics and visited exhibitions. I created illustrations and presentations in Canva and Crello.

**Trainer of Non-Formal Education at EuroCamp and Sources of Tolerance**
The Congress of National Communities of Ukraine — 2017—2020
I worked as a trainer with children aged 11-17 in an educational training course for teenagers. I created teams from teenagers who did not know each other and transformed them into one family using team-building exercises, progressing through all key stages of team development.

**Education**
**IT School Beetroot Academy**
Project Management in IT — 11/2020 - 03/2021

**Professional Experience**
**Project Manager in IT**
Epic Story — April 2021 - Present

### *Modified GPT-4o Resume Text (Not Formatted, PM Score 81.882)*

**Professional Summary**

Skilled Project Manager with expertise in document management for various project stages, creating prototypes, and selecting appropriate project methodologies. Hands-on experience with BPMN diagrams and adept in utilizing different project management tools, with a preference for ClickUp.

**Professional Experience**
**Project Manager in IT**

Epic Story — April 2021 - Present

Maintain comprehensive documentation across multiple stages of the project lifecycle, including Brief, Assumptions, WBS (WBS scheme, TTM, Calendar plan), Project Charter, MVP, Requirement Matrix, Risk Register, SRS, RACI, and Status Reports. Create project prototypes and determine the most effective project management methodologies. Design and implement BPMN diagrams to streamline project workflows. Utilize various PM tools for efficient project execution, with a specific focus on ClickUp.

**Trainer of Non-Formal Education**

EuroCamp and Sources of Tolerance, The Congress of National Communities of Ukraine — 2017—2020

Conducted educational training courses for children aged 11-17, focusing on building cohesive teams from diverse groups of teenagers. Facilitated team development through team-building activities, ensuring each group progressed through all key stages of team growth.

**Coordinator of Club of Tolerance**

The Congress of National Communities of Ukraine — 2017—2019

Organized and led weekly meetings with teenagers, engaging them in discussions on various topics and organizing visits to exhibitions. Created visual content and presentations using Canva and Crello to enhance meeting engagement and learning.

**Education**
**IT School Beetroot Academy**

Certification in Project Management in IT — 11/2020 - 03/2021

**Key Skills**
Document Management
Project Prototyping
Project Methodology Selection
BPMN Diagram Creation
ClickUp Proficiency
Team Building & Development
Non-Formal Education Training
Visual Content Creation (Canva, Crello)

In sum, as the lower and higher-scoring resumes were distinguishable by inspection in our experiments, we find qualitative evidence to support our "best-responding" candidates assumption.

# D. Experiment Details

### D.1. Dataset Details

The dataset we used, the Djinni Recruiting dataset, uses the MIT Licence and adheres to the conditions of fair use (Drushchak & Romanyshyn, 2024).

### D.2. Model Details

We include the costs of the different models we used. We used Together.ai to query several other models. According to pricing, CLAUDE-3.5-SONNET and GPT-4O would be the premium models both for input and output tokens. Other cost-efficient models that perform relative-well for improving resume scores are LLAMA3.3-70B-INSTRUCT-TURBO and DEEPSEEK-68B. We note that these models (with the exception of DeepSeek recently) are not broadly available to consumers.

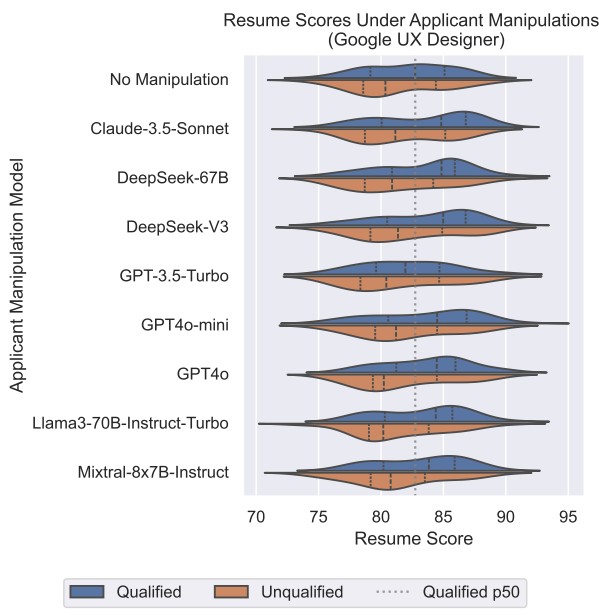

*Figure 10.* Resume score distribution of 50 qualified and 50 unqualified resumes before and after LLM manipulations for a Google UX Designer Position

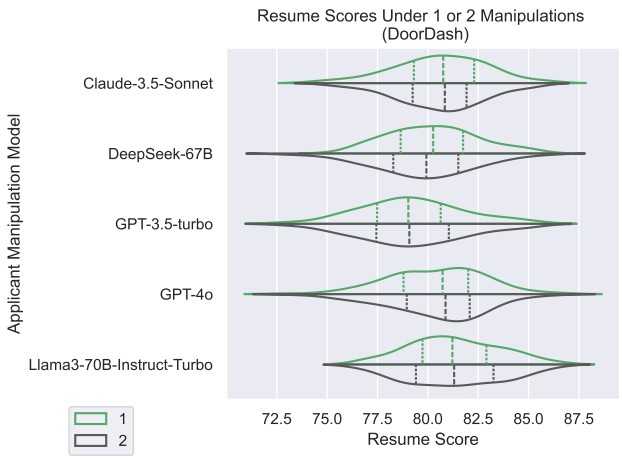

*Figure 11.* Applying LLM manipulations twice did not significantly improve the score of a resume more than a single manipulation.

| Model | Platfrom | Price (1M Input Tokens) | Price (1M Output Tokens) | Our Spend |
|---|---|---|---|---|
| Claude-3.5-Sonnet | Claude.ai | $1.50 | $7.50 | $1.04 |
| DeepSeek-67B | Together.ai | $0.90 | $0.90 | $0.29 |
| DeepSeek-V3 | DeepSeek | $0.55 | $2.19 | $0.11 |
| GPT-3.5-Turbo-0125 | OpenAI | $0.25 | $0.50 | $0.28 |
| GPT-4o-mini | OpenAI | $0.075 | $0.30 | $0.42 |
| GPT-4o-2024-08-06 | OpenAI | $1.25 | $5.00 | $80.04 |
| Mixtral-8x7b-Instruct | Together.ai | $0.60 | $0.60 | $7.44 |
| Llama3.3-70B-Instruct-Turbo | Together.ai | $0.88 | $0.88 | $0.73 |

*Table 7.* Summary of all models we experimented with. For transparency, we also include the total amount we spent on each model. Costs for GPT-4o and Mixtral-8x7b-Instruct are larger due to inital experiments.

### D.3. LLM Prompt Details

Preliminary testing with LLMs showed that they were easily susceptible to hallucinations. For instance, when we prompted the model with our *job-specific LLM prompt* (described further below), it frequently fabricated details about project management tools and methodologies that the candidate had not mentioned in their original resume. To empirically assess the susceptibility of LLMs to hallucinations, we tested resume modifications using a prompt designed to improve resumes based on a specific job description. The job description used in our experiments, shown below, was drawn from an example Project Manager role in the Djinni dataset.(Drushchak & Romanyshyn, 2024):

***Job-Specific LLM Prompt*:**

"Can you tailor my resume to this job description?

'A commitment to collaborative problem solving, agile thinking, and adaptability is essential. We are looking for a candidate who is able to balance a fast moving and changing environment with the ability to identify, investigate, and predict project risks Recruiting stages: HR interview, Tech interview **Core Responsibilities:** - Manage the full project life cycle including requirements gathering, creation of project plans and schedules, obtaining and managing resources, and facilitating project execution, deployment, and closure. - In cooperation with Technical Leads create and maintain comprehensive project documentation. - Manage Client expectations, monitor and increase CSAT level; - Plan, perform and implement process improvement initiatives. - Organize, lead, and facilitate cross-functional project teams. - Prepare weekly and monthly project status reports **What you need to Succeed:** - 1+ Year of dedicated Project Management in a production environment - Excellent organization and communication skills and the ability to communicate effectively with customers and co-workers. - Strong understanding of a Project Management Methodology (SDLC, Agile, Waterfall, etc.) - Creative mind with the ability to think outside-of-the-box. - The ability to manage multiple projects simultaneously - Experience with Jira or similar project management tool - Upper-intermediate level of English is a must.'

Please wait for me to enter my resume."

We then tested the aforementioned *Job-Specific LLM Prompt* with CHATGPT-4O on the following resume of a UI/UX designer:

**Original Input Resume Text:**

Possession of Adobe Photoshop, Adobe-AI, skills of working with a graphic tablet; UI/UX ; Figma, Zeplin, Sketch, MarvelApp, Invision; 2D graphics (understanding of pixel and vector graphics and their specifics); 2D Design; 3D Max; Ability to work in different styles; Ability to draw by hand; Creation of design banners, layouts, logos, frameworks and prototypes for websites and mobile applications;Testing of games; Making presentations in Power Point; Availability of portfolio; Bug tracking systems and project management tools: Trello and Jira. I'm self-motivated artist with a strong skills at game industry. I have more 5 years experience in game company, and now, I want to broaden my world with interesting projects, to share own experience and to acquire new skills and knowledge. So, my goal is putting into practice somebody's dreams, in so doing, working with pleasure. I would like to work at company with friendly team, interesting projects and a decent wage.

***Job-Specific LLM Prompt* Modified Resume:**

Your Name
Your Address
City, State, Zip
Your Email
Your Phone Number
LinkedIn Profile or Portfolio Link

*Professional Summary*

Self-motivated project manager with over 5 years of experience in the game industry, specializing in UI/UX design and project management. Adept at managing the full project life cycle, fostering collaboration, and implementing process improvements. Seeking to leverage strong organizational and communication skills in a dynamic environment.

*Core Competencies*

- *Project Management Methodologies:* Agile, Waterfall, SDLC
- *Tools:* Jira, Trello, Figma, Adobe Photoshop, Adobe Illustrator
- *Communication & Collaboration:* Excellent ability to manage client expectations and facilitate cross-functional teams
- *Process Improvement:* Proven track record in planning and implementing initiatives that enhance efficiency
- *Risk Management:* Skilled in identifying, investigating, and predicting project risks
- *Creative Problem Solving:* Ability to think outside the box and balance multiple projects simultaneously

*Professional Experience*

*Your Current Job Title*

Your Current Company Name, City, State                                      Month, Year to Present

Managed the full project life cycle, from requirements gathering to project execution and closure, for multiple UI/UX design projects.
Collaborated with technical leads to create and maintain comprehensive project documentation, ensuring clarity and alignment throughout the project.
Organized and led cross-functional project teams to facilitate successful project delivery, enhancing customer satisfaction and increasing CSAT levels.
Prepared and presented weekly and monthly project status reports to stakeholders, ensuring transparency and effective communication.

*UI/UX Designer*

Previous Company Name, City, State                                      Month, Year to Month, Year

Created design banners, layouts, logos, frameworks, and prototypes for websites and mobile applications, contributing to the overall project goals.
Conducted testing for game projects, utilizing bug tracking systems to enhance project quality and performance.
Developed and delivered engaging presentations in PowerPoint to communicate design concepts and project updates.

*Education*

*Your Degree*

Your University Name, City, State                                      Year of Graduation

*Portfolio*

Available upon request or at *link*.

*Languages*

English (Upper-intermediate)

*Interests*

Collaborative design projects
Continuous learning and skill development

As noted above, CHATGPT-4O fabricated details about the individual's technical project management skills. Specifically, it borrowed language (highlighted in blue) from the input prompt, which referenced the job description, and then inserted it into the outputted resume. These skills were mentioned in the job description but were not included in the individual's original resume.

D.3.1. FINAL ANTI-HALLUCINATION LLM PROMPT

To mitigate hallucinatiomns, we used he following prompt for our experiments. Additionally, we found that a sequential messaging style (without interruptions from the LLM) effectively reinforced the importance of preventing hallucinations to the LLM.

*Table 8.* Anti-Hallucination LLM Prompt

| Message No. | Text |
| --- | --- |
| 1. | "Can you improve my resume?" |
| 2. | "I will be punished if I lie about my qualifications, so it is imperative that the new resume does not add any facts that are not in the original resume." |
| 3. | "Also, please do not include any additional notes or explanation: I just want the text of the new resume." |
| 4. | "This is my resume: [*input resume*]." |

***Anti-Hallucination Prompt* Output Resume:**

*Your Name*
*Contact Information:*
*Professional Summary*
Self-motivated artist with over 5 years of experience in the game industry. Skilled in graphic design and UI/UX, I am eager to broaden my horizons through engaging projects, sharing my expertise, and acquiring new skills.

*Skills*

- Proficient in Adobe Photoshop and Adobe Illustrator
- UI/UX Design
- Experienced with Figma, Zeplin, Sketch, MarvelApp, and InVision
- Understanding of pixel and vector graphics
- 2D and 3D Design (3D Max)
- Hand-drawing capabilities
- Creation of design banners, layouts, logos, frameworks, and prototypes for websites and mobile applications
- Game testing experience
- Proficient in creating presentations in PowerPoint
- Familiarity with bug tracking systems and project management tools: Trello and Jira

*Portfolio*
Available upon request.

*Objective*
To contribute to exciting projects within a collaborative team environment while achieving a balanced professional development and fair compensation.

We also tested this *Anti-Hallucination Prompt* on our selected test resume (see "Original Input Resume Text" on the previous page): the subsequent outputted resume is shown above. When compared to the output of our *Job-Specific LLM Prompt*, we found that our *Anti-Hallucination Prompt* successfully did not fabricate information that was not previously mentioned in the original resume. Instead, it solely improved the written quality of resumes by adding a "professional summary section" and reorganizing the individual's existing skills in a logical manner. Upon manual inspection of a random subset of outputs from our *Anti-Hallucination Prompt*, we further confirmed that hallucinations were indeed reduced by this prompt.

### D.4. Evaluation (Scoring) of Resumes

To evaluate our (potentially LLM-manipulated) resumes, we used Resume-Matcher to compare them against two Project Manager job descriptions, which we selected from job boards on LinkedIn (DoorDash Inc., n.d.) and Google (Google, n.d.). We selected these two descriptions due to their detailed explanation of their respective role-related responsibilities and their lexical dissimilarity to each other.

*Job Description 1 (Product Manager (Multiple Levels) @ DoorDash)***:**

> **About the Team:** At DoorDash, we're redefining the future of on-demand delivery. To do this, we're building a world-class product organization, in which each of our product managers plays a critical role in helping to define and execute our vision to connect local delivery networks in cities all across the world.
>
> **About The Role:** Product Managers at DoorDash require a sharp consumer-first eye, platform thinking, and strong cross-functional collaboration. As a Product Manager at DoorDash, you will own the product strategy and vision, define the product roadmap and alignment, and help drive the execution. You will be working on mission-critical products that shape the direction of the company. You will report into one of the following pillars: Merchant, Consumer, Operational Excellence, Ads, Logistics, or New Verticals. This role is a hybrid of remote work and in-person collaboration.
>
> **You're Excited About This Opportunity Because You Will...**
>
> 1. Drive the product definition, strategy, and long-term vision. You own the roadmap.
> 2. Work closely with cross-functional teams of designers, operators, data scientists, and engineers.
> 3. Communicate product plans, benefits, and results to key stakeholders, including the leadership team.
>
> **We're Excited About You Because...**
>
> 1. You have 5+ years of Product Management industry experience.
> 2. You have 4+ years of user-facing experience in industries such as eCommerce, technology, or multi-sided marketplaces.
> 3. You have proven abilities in driving product strategy, vision, and roadmap alignment.
> 4. You're an execution powerhouse.
> 5. You have experience presenting business reviews to senior executives.
> 6. You have empathy for the users you build for.
> 7. You are passionate about DoorDash and the problems we are solving for.
>
> **About DoorDash:** At DoorDash, our mission to empower local economies shapes how our team members move quickly, learn, and reiterate in order to make impactful decisions that display empathy for our range of users—from Dashers to merchant partners to consumers. We are a technology and logistics company that started with door-to-door delivery, and we are looking for team members who can help us go from a company that is known for delivering food to a company that people turn to for any and all goods. DoorDash is growing rapidly and changing constantly, which gives our team members the opportunity to share their unique perspectives, solve new challenges, and own their careers. We're committed to supporting employees' happiness, healthiness, and overall well-being by providing comprehensive benefits and perks including premium healthcare, wellness expense reimbursement, paid parental leave, and more.
>
> **Our Commitment to Diversity and Inclusion:** We're committed to growing and empowering a more inclusive community within our company, industry, and cities. That's why we hire and cultivate diverse teams of people from all backgrounds, experiences, and perspectives. We believe that true innovation happens when everyone has room at the table and the tools, resources, and opportunity to excel.

**Statement of Non-Discrimination:** In keeping with our beliefs and goals, no employee or applicant will face discrimination or harassment based on: race, color, ancestry, national origin, religion, age, gender, marital/domestic partner status, sexual orientation, gender identity or expression, disability status, or veteran status. Above and beyond discrimination and harassment based on 'protected categories,' we also strive to prevent other subtler forms of inappropriate behavior (i.e., stereotyping) from ever gaining a foothold in our office. Whether blatant or hidden, barriers to success have no place at DoorDash. We value a diverse workforce – people who identify as women, non-binary or gender non-conforming, LGBTQIA+, American Indian or Native Alaskan, Black or African American, Hispanic or Latinx, Native Hawaiian or Other Pacific Islander, differently-abled, caretakers and parents, and veterans are strongly encouraged to apply. Thank you to the Level Playing Field Institute for this statement of non-discrimination.

Pursuant to the San Francisco Fair Chance Ordinance, Los Angeles Fair Chance Initiative for Hiring Ordinance, and any other state or local hiring regulations, we will consider for employment any qualified applicant, including those with arrest and conviction records, in a manner consistent with the applicable regulation. If you need any accommodations, please inform your recruiting contact upon initial connection.

### *Job Description 2 (Staff UX Designer @ Google)*

**Minimum Qualifications**

1. Bachelor's degree in Design, Human-Computer Interaction, Computer Science, a related field, or equivalent practical experience.
2. 8 years of experience in product design or UX.
3. Experience with industry standard design tools (e.g., Photoshop, Illustrator, Sketch, InVisio, Figma, Principle, etc.).
4. Include a portfolio, website, or any other relevant link to your work in your resume (providing a viewable link or access instructions).

**Preferred Qualifications:**

1. Experience creating and maintaining robust, coherent design systems that are usable across various devices and platforms.
2. Experience shaping processes to establish and sustain a scalable and consistent design language.
3. Proven track record of exploring and implementing innovative communication modes that are delightful, understandable, and accessible for diverse user groups, including designers, developers, and end-users.
4. Demonstrated ability to provide a strong design perspective with fluency in native app patterns and user experiences.
5. Strong collaboration skills within cross-functional product teams, including working with engineering, writers, editors, and researchers to refine and validate design decisions.

**About the Job:** At Google, we follow a simple but vital premise: "Focus on the user and all else will follow." Google's Interaction Designers take complex tasks and make them intuitive and easy-to-use for billions of people around the globe. Throughout the design process—from creating user flows and wireframes to building user interface mockups and prototypes—you'll envision how people will experience our products, and bring that vision to life in a way that feels inspired, refined, and even magical.

Google User Experience (UX) is made up of multi-disciplinary teams of UX Designers, Researchers, Writers, Content Strategists, Program Managers, and Engineers: we care deeply about the people who use our products. The UX team plays an integral part in gathering insights about the attitudes, emotions, and behaviors of people who use our products to inspire and inform design. We collaborate closely with each other and with engineering and product management to create industry-leading products that deliver value for the people who use them, and for Google's businesses.

As an Interaction Designer, you'll rely on user-centered design methods to craft industry-leading user experiences—from concept to execution. Like all of our UX jobs, you'll collaborate with your design partners to leverage and evolve the Google design language to build beautiful, innovative, inspired products that people love to use.

Labs is a group focused on incubating early-stage efforts in support of Google's mission to organize the world's information and make it universally accessible and useful. Our team exists to help discover and create new ways to advance our core products through exploration and the application of new technologies. We work to build new solutions that have the potential to transform how users interact with Google. Our goal is to drive innovation by developing new Google products and capabilities that deliver significant impact over longer timeframes.

The US base salary range for this full-time position is $168,000-252,000$ + bonus + equity + benefits. Our salary ranges are determined by role, level, and location. The range displayed on each job posting reflects the minimum and maximum target salaries for the position across all US locations. Within the range, individual pay is determined by work location and additional factors, including job-related skills, experience, and relevant education or training. Your recruiter can share more about the specific salary range for your preferred location during the hiring process.

Please note that the compensation details listed in US role postings reflect the base salary only, and do not include bonus, equity, or benefits. Learn more about benefits at Google.

**Responsibilities:** Influence cross-functional stakeholders to gain support for design strategies, collaborating from early-stage concept development to iteration and execution. Drive the creation of innovative design solutions that address user needs, business objectives, and industry trends while furthering business outcomes. Build and iterate on Figma prototypes to communicate ideas, user journeys, and decision points, while exploring rapid visual design styles and interactions. Apply user-centered design principles, integrating UX research insights and AI-first interactions to create unique, multi-modal user experiences that drive product adoption. Provide direction to UX designers, manage priorities, dependencies, and stakeholders effectively, and independently drive work toward key milestones.

## E. Examples

We provide a simple example to illustrate the intuition behind the TWO-TICKET scheme.

**Example 1.** Consider some joint distribution $\mathcal{D}$ over feature vectors, groups, and labels, as well some scoring function $s$ such that $s(\boldsymbol{X}) \sim \text{Uni}([0, 10])$ and $(H \mid \boldsymbol{X}) = \mathbb{1}[s(\boldsymbol{X}) \geq 5]$.

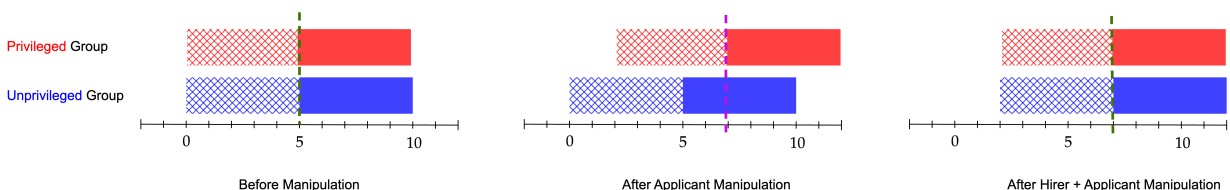

*Figure 12.* Example resume score distributions for Group $P$ and Group $U$. Solid regions correspond to truly represent candidates, while the hatched regions represent truly unqualified candidates.

1. If the Hirer were to receive the unmanipulated scores, then the threshold $\tau = 5$ would achieve perfect classification for both groups, as shown in the leftmost plot.

2. However, suppose that the groups have unequal access to LLMs. Namely, suppose Group $P$ has access to an LLM $L_P$ that increases scores by 1, while $U$ has no access to LLMs. Since the Hirer cannot discriminate between scores from different groups differently, no threshold can simultaneously classify both groups correctly. The interval of undominated thresholds is $[5, 6]$: however, any threshold in this range must accept unqualified candidates from Group $P$ or reject qualified candidates from Group $U$. This harms both fairness and overall accuracy. Under the No False Positives Objective, the optimal threshold is $\tau^* = 6$ with $\text{TPR}_P = 1$ but $\text{TPR}_U = 0.8$, as shown in the middle plot.

3. To mitigate this disparity, consider applying Hirer LLM $L_H$ where $L_H(L_P(s(\boldsymbol{x}))) = s(\boldsymbol{x}) + 1$ and $L_H(L_U(s(\boldsymbol{x}))) = s(\boldsymbol{x}) + 1$. The idea is that $L_H$ improves Group $U$'s scores while minimally affecting Group $P$'s scores, restoring equality of post-manipulation score distributions between the two groups. This allows for perfect classification between the two Groups again with a new threshold $\tau^* = 6$, as shown in the rightmost plot.

# F. Complete Proofs

**Theorem 1.** Suppose $L_P \succeq L_U$. Then for all $\boldsymbol{x} \in \mathbb{R}^d$, $\Delta(\boldsymbol{x}) \geq 0$.

*Proof.* Fix $\boldsymbol{x} \in \mathcal{X}$. If $f_\tau(\boldsymbol{x}) = 1$, then trivially $\Delta(\boldsymbol{x}) = 0$. We therefore focus on the case that $f_\tau(\boldsymbol{x}) = 0$. In this case, note that

$$\mathbb{P}_{L_g}(f_\tau(\boldsymbol{x}'_g) = 1) = \mathbb{E}_{L_g}[f_\tau(L_g(\boldsymbol{x}))].$$

Since $s$ is non-decreasing, $f_\tau(\boldsymbol{x}) = \mathbb{1}\left[s(\boldsymbol{x}) \geq \tau\right]$ is also non-decreasing. Since $L_P \succeq L_U$, $L_P(\boldsymbol{x})$ stochastically dominates $L_U(\boldsymbol{x})$, so by Lemma 5.1,

$$\mathbb{E}_{L_g}[f_\tau(L_P(\boldsymbol{x}))] \geq \mathbb{E}_{L_g}[f_\tau(L_U(\boldsymbol{x}))]$$

$$\implies \mathbb{P}_{L_g}(f_\tau(\boldsymbol{x}'_P) = 1) \geq \mathbb{P}_{L_g}(f_\tau(\boldsymbol{x}'_U) = 1)$$

$$\implies \mathbb{P}_{L_g}(f_\tau(\boldsymbol{x}'_P) = 1) - \mathbb{P}_{L_g}(f_\tau(\boldsymbol{x}'_U) = 1) \geq 0.$$

$$\implies \Delta(\boldsymbol{x}) \geq 0. \square$$

**Corollary 1.** Suppose $L_P \succeq L_U$. Then, $\Delta_{\mathrm{TPR}} \geq 0$.

*Proof.* Observe that $\Delta_{\mathrm{TPR}} = \mathbb{E}_{\boldsymbol{x}}[\Delta(\boldsymbol{x}) \mid Y = 1]$. Since $L_P \succeq L_U$, Theorem 1 implies that for all $\boldsymbol{x} \in \mathcal{X}$, $\Delta(\boldsymbol{x}) \geq 0$. Thus, $\Delta_{\mathrm{TPR}} \geq 0$.

$\square$

**Lemma 6.1.** If $L_P \succeq L_H^{(1)}, L_H^{(2)}$, then $\tau^{*(1)} = \tau^{*(2)}$.

*Proof.* Assuming a continuous distribution over scores, the minimum threshold that achieves the No False Positives Objective is equal to the maximum score achievable by a candidate with true label $Y = 0$. That is,=

$$\tau^{*(k)} = \max s(\boldsymbol{x}'')$$

where

$$\boldsymbol{x}'' \in \{\boldsymbol{x}', L_H^{(k)}(\boldsymbol{x}')\}$$
$$\text{and } \boldsymbol{x}' \in \{\boldsymbol{x}, L_P(\boldsymbol{x}, L_U(\boldsymbol{x}))\}$$
$$\text{and } \boldsymbol{x} \sim \mathcal{D} \mid Y = 0.$$

By leveraging the conditional independence $L_H(\boldsymbol{x}'_g)$ and $\boldsymbol{x}'_g$ given $\boldsymbol{x}$ (the same logic as the proof of Lemma 6.2), we may equivalently write

$$\tau^{*(k)} = \max\left(M, M_P, M_U, M_H^{(k)}\right).$$

where

$$M := \max_{x \sim \mathcal{D} \mid Y = 0} s(\boldsymbol{x})$$

$$M_g := \max_{L_g, x \sim \mathcal{D} \mid Y = 0} s(L_g(\boldsymbol{x}))$$

$$M_H^{(k)} := \max_{L_H^{(k)}, x \sim \mathcal{D} \mid Y = 0} s(L_H^{(k)}(\boldsymbol{x})),$$

We show that, under the Lemma's condition, $M_H^{(k)}$ is irrelevant to this expression. Suppose $L_P \succeq L_H^{(k)}$. Then, fixing $\boldsymbol{x} \in \mathcal{X}$, $L_P(\boldsymbol{x})$ stochastically dominates $L_H^{(k)}(\boldsymbol{x})$. Since $s$ is non-decreasing, $s(L_P(\boldsymbol{x}))$ stochastically dominates $s(L_H^{(k)}(\boldsymbol{x}))$. Thus, $\max_{L_P} s(L_P(\boldsymbol{x})) \geq \max_{L_H^{(k)}} s(L_H^{(k)}(\boldsymbol{x}))$. Taking the maximum over $\boldsymbol{x} \sim \mathcal{D} \mid Y = 0$ yields

$$M_P = \max_{L_P, x \sim \mathcal{D} \mid Y = 0} s(L_P(\boldsymbol{x})) \geq \max_{L_H^{(k)}, x \sim \mathcal{D} \mid Y = 0} s(L_H^{(k)}(\boldsymbol{x})) = M_H^{(k)}.$$

Since $M_P \geq M_H^{(k)}$, we have $\tau^{*(k)} = \max(M, M_P, M_U)$. This expression is independent of $k$, so $\tau^{*(1)} = \tau^{*(2)}$.

$\square$

**Lemma 6.2.** For Hirer LLM $L_H$ and threshold $\tau$, the probability that a candidate $(\boldsymbol{x}, g, y)$ is accepted is

$$\mathbb{P}_{L_g, L_H} (f_\tau(\boldsymbol{x}_g'') = 1) =$$
$$1 - \mathbb{1}\left[s(\boldsymbol{x}) < \tau\right] \cdot \mathbb{P}_{L_g} \left(s(L_g(\boldsymbol{x})) < \tau\right) \cdot \mathbb{P}_{L_H} \left(s(L_H(\boldsymbol{x})) < \tau\right).$$

*Proof.* If $s(\boldsymbol{x}) \geq \tau$, then trivially $\mathbb{P}_{L_g, L_H}(f_\tau(\boldsymbol{x}_g'') = 1) = 1$. We therefore focus on the case that $f_\tau(\boldsymbol{x}) = 0$. In this case, note that

$$\mathbb{P}_{L_g, L_H} (f_\tau(\boldsymbol{x}_g'') = 0) = \mathbb{P}_{L_g, L_H} \left(s(\boldsymbol{x}_g'') < \tau\right) = \mathbb{P}_{L_g, L_H} \left(s(\boldsymbol{x}_g') < \tau \cap s(L_H(\boldsymbol{x}_g')) < \tau\right).$$

Observe from the definition of LLM manipulation that $L_H(\boldsymbol{x}_g')$ and $\boldsymbol{x}_g'$ are conditionally independent given $\boldsymbol{x}$. Thus,

$$\mathbb{P}_{L_g, L_H} (f_\tau(\boldsymbol{x}_g'') = 0) = \mathbb{P}_{L_g} \left(s(\boldsymbol{x}_g') < \tau\right) \mathbb{P}_{L_g, L_H} \left(s(L_H(\boldsymbol{x}_g')) < \tau\right).$$

Furthermore, observe that $L_H(\boldsymbol{x}_g')$ is equal in distribution to $L_H(\boldsymbol{x})$. We obtain

$$\mathbb{P}_{L_g, L_H} (f_\tau(\boldsymbol{x}_g'') = 0) = \mathbb{P}_{L_g} \left(s(L_g(\boldsymbol{x}_g)) < \tau\right) \mathbb{P}_{L_H} \left(s(L_H(\boldsymbol{x}_g)) < \tau\right).$$

Taking the complement yields the lemma. $\square$

**Theorem 2.** Let $\tau^{*(1)} = \tau^{*(2)}$, $L_P \succeq L_U$, and $L_H^{(2)} \succeq L_H^{(1)}$. Then for all $\boldsymbol{x} \in \mathbb{R}^d$, $\Delta^{(2)}(\boldsymbol{x}) \leq \Delta^{(1)}(\boldsymbol{x})$.

*Proof.* For convenience, let $\tau = \tau^{*(1)} = \tau^{*(2)}$ be the common threshold that achieves the No False Positives Objective. Fix $\boldsymbol{x} \in \mathcal{X}$. If $f_\tau^{(1)}(\boldsymbol{x}) = f_\tau^{(2)}(\boldsymbol{x}) = 1$, then trivially $\Delta^{(1)}(\boldsymbol{x}) = \Delta^{(2)}(\boldsymbol{x}) = 0$. We therefore focus on the case that $f_\tau^{(1)}(\boldsymbol{x}) = f_\tau^{(2)}(\boldsymbol{x}) = 0$. By Lemma 6.2,

$$\Delta^{(k)}(\boldsymbol{x}) = \mathbb{P}_{L_P, L_H^{(k)}}(f_\tau(\boldsymbol{x}_P'') = 1) - \mathbb{P}_{L_U, L_H^{(k)}}(f_\tau(\boldsymbol{x}_U'') = 1)$$
$$= \mathbb{P}_{L_H^{(k)}}\left(s(L_H^{(k)}(\boldsymbol{x})) < \tau\right) \cdot d(\boldsymbol{x}),$$

where $d(\boldsymbol{x}) = \mathbb{P}_{L_U}\left(s(L_U(\boldsymbol{x}) < \tau\right) - \mathbb{P}_{L_P}\left(s(L_P(\boldsymbol{x}) < \tau\right)$.

Observe that $d(\boldsymbol{x})$ does not depend on the Hiring Scheme $k$. Thus,

$$\Delta^{(2)}(\boldsymbol{x}) - \Delta^{(1)}(\boldsymbol{x}) = -\delta(\boldsymbol{x}) \cdot d(\boldsymbol{x}),$$

where $\delta(\boldsymbol{x}) = \mathbb{P}_{L_H^{(1)}}(s(L_H^{(1)}(\boldsymbol{x})) < \tau) - \mathbb{P}_{L_H^{(2)}}(s(L_H^{(2)}(\boldsymbol{x})) < \tau)$.

Since $L_P \succeq L_U$ by assumption, by Theorem 1, $d(\boldsymbol{x}) \geq 0$. By a very similar argument, since $L_H^{(2)} \succeq L_H^{(1)}$, $\delta(\boldsymbol{x}) \geq 0$. It follows that

$$\Delta^{(2)}(\boldsymbol{x}) - \Delta^{(1)}(\boldsymbol{x}) \leq 0.$$

$\square$

**Corollary 2.** Let $\text{TPR}_g^{(k)}$ denote the true positive rate over group $g$ under Hiring Scheme $k$. Let $\tau^{*(1)} = \tau^{*(2)}$, $L_P \succeq L_U$, and $L_H^{(2)} \succeq L_H^{(1)}$. Then

1. $\left|\Delta_{\text{TPR}}^{(2)}\right| \leq \left|\Delta_{\text{TPR}}^{(1)}\right|$.

2. $\text{TPR}_g^{(2)} \geq \text{TPR}_g^{(1)}$ for $g \in \{P, U\}$.

3. $\text{TPR}^{(2)} \geq \text{TPR}^{(1)}$.

*Proof.* The first part follows almost immediately from Theorem 2 upon observing that

$$\Delta_{\text{TPR}} = \mathop{\mathbb{E}}_{L_P, L_U, \boldsymbol{x}} \left[ \Delta(\boldsymbol{x}) \mid Y = 1 \right].$$

The second part follows from an application of Lemma 6.2 and a near-identical argument to Theorem 2.

The third part follows from the second part, observing that $\text{TPR}^{(k)} = \mathbb{P}(G = P)\text{TPR}_P^{(k)} + \mathbb{P}(G = U)\text{TPR}_U^{(k)}$. $\qquad \square$

**Theorem 4.** Let $\tau$ be the threshold used by the Hirer in the $n$-ticket scheme. If $L_H \succeq L_P \succeq L_U$, applying the $n$-ticket scheme and taking the limit as $n \to \infty$, then any applicant $\boldsymbol{x} \in \mathcal{D}$ is guaranteed to receive a group-independent outcome, $o = o(\boldsymbol{x}, L_H) \in \{0, 1\}$. Furthermore, there exists $k_{\boldsymbol{x}} \in [0, 1)$, dependent on $L_H$ and $\boldsymbol{x}$, such that

$$|\mathbb{P}(f_\tau(L_H^n(L_g(\boldsymbol{x}))) = 1) - o| \leq \mathcal{O}(k_{\boldsymbol{x}}^n).$$

We prove the theorem in two different ways. The first is a relatively straightforward argument based on Lemma 6.2. In the second, we show that the acceptance mapping induced by the Hirer's LLM is a contraction operator, allowing us to apply Banach's Fixed Point Theorem. We include the fixed point proof as it is standard in similar settings and provides useful intuition about the role of the Hirer's LLM.

*Proof.* By similar logic to Lemma 6.2, under an $n$-ticket scheme, we have

$$\mathop{\mathbb{P}}_{L_g, L_H} (f_\tau(\boldsymbol{x}^{(n)}) = 1) = 1 - \mathbb{1}\left[s(\boldsymbol{x}) < \tau\right] \cdot \mathop{\mathbb{P}}_{L_g} \left(s(L_g(\boldsymbol{x})) < \tau\right) \cdot \mathop{\mathbb{P}}_{L_H} \left(s(L_H(\boldsymbol{x})) < \tau\right)^{n-1}.$$

If $s(\boldsymbol{x}) \geq \tau$, then $\mathbb{P}_{L_g, L_H}(f_\tau(\boldsymbol{x}^{(n)}) = 1) = 1$ for all $n \geq 1$.

If $s(\boldsymbol{x}) < \tau$, let $k_{\boldsymbol{x}} = \mathbb{P}_{L_H}\left(s(L_H(\boldsymbol{x})) < \tau\right)$. Then we have

$$\mathop{\mathbb{P}}_{L_g, L_H} (f_\tau(\boldsymbol{x}^{(n)}) = 1) = 1 - \mathop{\mathbb{P}}_{L_g} \left(s(L_g(\boldsymbol{x})) < \tau\right) \cdot k_{\boldsymbol{x}}^{n-1}.$$

If $k_{\boldsymbol{x}} = 1$, then since $L_H \succeq L_g$, we have $\mathbb{P}_{L_g}\left(s(L_g(\boldsymbol{x})) < \tau\right) \geq \mathbb{P}_{L_H}\left(s(L_H(\boldsymbol{x})) < \tau\right) = 1$. Thus, $\mathbb{P}_{L_g, L_H}(f_\tau(\boldsymbol{x}^{(n)}) = 1) = 0$ for all $n \geq 1$. If $0 \leq k_{\boldsymbol{x}} < 1$, then we have instead $\mathbb{P}(f_\tau(\boldsymbol{x}^{(n)}) = 1) \geq 1 - \mathcal{O}(k_{\boldsymbol{x}}^n)$. $\qquad \square$

*Proof.* Let $\boldsymbol{x} \in \mathbb{R}^d$ be an unmodified resume. For $g \in \{p, u\}$, let $g_{\boldsymbol{x}} = \mathbb{P}_{L_G}(f_\tau(L_g(\boldsymbol{x})) = 1)$ be the baseline, group-dependent probability of acceptance. Namely, $u_{\boldsymbol{x}}$ represents the probability that a candidate from group $U$ with unmodified resume $\boldsymbol{x}$ will be accepted by $f_\tau$ given their resume has been modified once by LLM $L_U$, with the same holding for $p_{\boldsymbol{x}}, P, L_P$, respectively.

Let $h_{\boldsymbol{x}} = \mathbb{P}_{L_H}(f_\tau(L_H(\boldsymbol{z})) = 1)$, where $\boldsymbol{z} \in \{L_H^i(L_G(\boldsymbol{x}))\}$ for $i \in \mathbb{N}$ and $G \in \{U, P\}$. Since any LLM manipulation is invariant to previous manipulations, $h_{\boldsymbol{x}}$ represents the probability that the result of a single $L_H$ application to a possibly manipulated resume $\boldsymbol{z}$ achieves a score equal to or above the threshold $\tau$. Observe that $g_{\boldsymbol{x}}$ and $h_{\boldsymbol{x}}$ depend on $\boldsymbol{x}$ due to the presence of fundamental features that are preserved throughout every LLM manipulation and affect the probability of acceptance.

Following Theorem 1 and the assumption in the theorem, we have that $u_{\boldsymbol{x}} \leq p_{\boldsymbol{x}} \leq h_{\boldsymbol{x}}$.

Consider the function $T_{\boldsymbol{x}} : [0, 1] \to [0, 1]$ given by

$$T_{\boldsymbol{x}}(z) = z + h_{\boldsymbol{x}}(1 - z).$$

Since $0 \leq z + h_{\boldsymbol{x}}(1 - z) \leq 1$, the output of $T_{\boldsymbol{x}}(z)$ is in $[0, 1]$ and thus $T$ is well-defined.

Let $d(p, u) = |p - u|$. Notice that $([0, 1], d)$ is a metric space. We will show that $T_{\boldsymbol{x}}$ is a contraction operator.

Let $k_{\boldsymbol{x}} = 1 - h_{\boldsymbol{x}}$. If $h_{\boldsymbol{x}} = 0$, the applicant will be rejected even after applying the two-ticket scheme an infinite number of times unless $f_\tau(\boldsymbol{x}) = 1$, in which case the applicant will be accepted for every $n \in \mathbb{N}$. Either way, if $h_{\boldsymbol{x}} = 0$, the outcome is independent of group membership.

So suppose $0 < h_{\boldsymbol{x}} < 1$. This implies $0 < k_{\boldsymbol{x}} < 1$. For any $p, u \in [0, 1]$, we have that

$$d(T_{\boldsymbol{x}}(p), T_{\boldsymbol{x}}(u)) = |p + h_{\boldsymbol{x}}(1 - p) - (u + h_{\boldsymbol{x}}(1 - u))| = (1 - h_{\boldsymbol{x}})|p - u| \leq k_{\boldsymbol{x}} d(T_{\boldsymbol{x}}(p), T_{\boldsymbol{x}}(u)).$$

Therefore, $T_{\boldsymbol{x}}$ is a contraction operator by definition.

Recall that we already showed that for $n \geq 2$, if $h_{\boldsymbol{x}} = 0$ then the outcome is independent of the group membership. Next, we show that for $h_{\boldsymbol{x}} > 0$, $T_{\boldsymbol{x}}^n(z)$ converges to 1, which implies that the outcome of the $n$-ticket scheme when $n \to \infty$ is always acceptance and is independent of group membership.

**Claim 5.** If $h_{\boldsymbol{x}} > 0$, $\lim_{n \to \infty} T_{\boldsymbol{x}}^n(z) = 1$.

*Proof.* Rearranging $T_{\boldsymbol{x}}(z)$,

$$T_{\boldsymbol{x}}(z) = (1 - h_{\boldsymbol{x}})z + h_{\boldsymbol{x}}.$$

Applying $T_{\boldsymbol{x}}$ twice:

$$T_{\boldsymbol{x}}^2(z) = (1 - h_{\boldsymbol{x}})((1 - h_{\boldsymbol{x}})z + h_{\boldsymbol{x}}) + h_{\boldsymbol{x}} = (1 - h_{\boldsymbol{x}})^2 z + h_{\boldsymbol{x}}(1 + (1 - h_{\boldsymbol{x}})).$$

Continuing this process, we can express $T_{\boldsymbol{x}}^n$ as a geometric series,

$$T_{\boldsymbol{x}}^n(z) = (1 - h_{\boldsymbol{x}})^n z + h_{\boldsymbol{x}} \sum_{j=0}^{n-1} (1 - h_{\boldsymbol{x}})^j = (1 - h_{\boldsymbol{x}})^n z + h_{\boldsymbol{x}} \cdot \frac{1 - (1 - h_{\boldsymbol{x}})^n}{h_{\boldsymbol{x}}} = 1 - (1 - h_{\boldsymbol{x}})^n(1 - z).$$

Taking the limit as $n \to \infty$:

$$\lim_{n \to \infty} T_{\boldsymbol{x}}^n(z) = 1 - \lim_{n \to \infty} (1 - h_{\boldsymbol{x}})^n(1 - z) = 1.$$

$\square$

The rest of the proof follows from Banach's fixed point theorem. $\square$

**Corollary 3.** If $L_H \succeq L_P \succeq L_U$, then for every unmodified resume $\boldsymbol{x} \in \mathbb{R}^d$, there exists $k_{\boldsymbol{x}} \in [0, 1)$ that depends on $L_H$ and $\boldsymbol{x}$ such that for any $n \geq 2$, $\mathbb{P}(f_\tau(L_H^n(L_U(\boldsymbol{x}))) \neq f_\tau(L_H^n(L_P(\boldsymbol{x}))) \leq O(k_{\boldsymbol{x}}^n)$. Hence,

1. $|\Delta_{TPR}^{(n)}| \leq \mathcal{O}(k^n)$, where $k = \max_{\boldsymbol{x}} k_{\boldsymbol{x}}$.
2. $\text{TPR}_g^{(n)} \geq \text{TPR}_g^{(n-1)}$ for $g \in \{P, U\}$ and $n > 1$.
3. $\text{TPR}^{(n)} \geq \text{TPR}^{(n-1)}$ for every $n > 1$.

Since the threshold $\tau^*$ already prevents false positives (Definition 4.2), (3) also implies that accuracy does not decrease.

*Proof.* If $o = 0$ then from Theorem 4, for any $G \in \{U, P\}$,

$$|\mathbb{P}(f_\tau(L_H^n(L_G(\boldsymbol{x}))) = 0) - 0| = \mathbb{P}(f_\tau(L_H^n(L_G(\boldsymbol{x}))) = 0) \leq O(k_{\boldsymbol{x}}^n).$$

If $o = 1$ then from Theorem 4, for any $G \in \{U, P\}$,

$$|\mathbb{P}(f_\tau(L_H^n(L_G(\boldsymbol{x}))) = 1) - 1| = 1 - \mathbb{P}(f_\tau(L_H^n(L_G(\boldsymbol{x}))) = 1) \leq O(k^n),$$

hence

$$1 - O(k_{\boldsymbol{x}}^n) \leq \mathbb{P}(f_\tau(L_H^n(L_G(\boldsymbol{x})) = 1).$$

Let $o = o(\boldsymbol{x})$ be the outcome of $\boldsymbol{x}$.

$$\mathbb{P}\left(f_\tau(L_H^n(L_U(\boldsymbol{x}))) \neq f_\tau(L_H^n(L_P(\boldsymbol{x}))\right) \leq \mathbb{P}(f_\tau(L_H^n(L_U(\boldsymbol{x}))) \neq o) \vee f_\tau(L_H^n(L_P(\boldsymbol{x}))) \neq o)$$

From union bound,

$$\mathbb{P}\left(f_\tau(L_H^n(L_U(\boldsymbol{x}))) \neq o) \vee f_\tau(L_H^n(L_P(\boldsymbol{x}))) \neq o\right) \leq 2\mathbb{P}(f_\tau(L_H^n(L_G(\boldsymbol{x})) \neq o) \leq 2O(k_{\boldsymbol{x}}^n) = O(k_{\boldsymbol{x}}^n).$$

As for the clauses, (1) follows directly from the above, by conditioning over $y = 1$. (2) and (3) follow from the same proof as Corollary 2. $\square$

