# OpenReview forum: "Two Tickets are Better than One: Fair and Accurate Hiring Under Strategic LLM Manipulations"
_ICML.cc/2025/Conference — ICML 2025 poster_

### Official Review · Reviewer_oTGu · 2025-03-07

**Overall Recommendation:** 3

**Summary:**

This paper studies a new variant of strategic classification applied to automated hiring decisions when applicants use large language models (LLMs) to improve (i.e., “manipulate”) their resumes. The authors observe that LLM-based enhancements can blur the line between skilled and unskilled applicants, especially when different applicants have access to different-quality models.
Through theoretical analysis and experiments on real resumes, the paper shows that this two-ticket approach can simultaneously improve both overall accuracy (true positive rate) and fairness (reduce disparities between applicants who have access to high-quality LLMs and those who do not).

**Claims And Evidence:**

1. Point 1
a. Claim: Traditional hiring pipelines are susceptible to unequal LLM access—applicants who can afford better models can often achieve substantially higher resume scores and thus enjoy higher acceptance rates.
b. Evidence: Empirical experiments on 520 real resumes demonstrate that using strong LLMs (like GPT-4 variants) can significantly increase a resume’s relevance scores—sometimes making an unqualified resume appear indistinguishable from a qualified one (as shown in Figure 1).

2. Point 2
a. Claim: A “two-ticket” approach can improve both overall accuracy and fairness. The hiring algorithm can re-manipulate the candidate’s final resume with the “two-ticket” approach.
b. Evidence: The authors present formal theorems (Theorem 2 and Corollary 2) and a set of experiments to show that, under a “no false positives” objective (prioritizing zero FPR and maximizing TPR), giving every candidate a second manipulation step (via the hiring algorithm’s own LLM) raises TPR for both privileged and unprivileged groups while reducing TPR disparity.

**Essential References Not Discussed:**

Potentially under-discussed areas that might be valuable:

1. Work in human-AI collaboration or human-in-the-loop hiring, which could complement or replace purely automated scoring.
2. Emerging “AI detection” strategies or watermarking of LLM output. The authors mention not knowing whether the text was manipulated but do not deeply discuss whether detection-based solutions might reduce disparities.

**Experimental Designs Or Analyses:**

1. Data: They use 520 resumes (balanced across UI/UX design and project management roles) from an open-source dataset that includes anonymized real resumes.
2. LLM Conditions: Different groups get different LLMs (e.g., GPT-3.5 vs. GPT-4 variants); the better model is assigned to the “privileged” group. Applicants submit whichever version of their resume scores higher (original or LLM-manipulated).
3. Two-Ticket Step: The hiring algorithm re-manipulates each submitted resume with its own GPT-4-based LLM, and the final acceptance decision is based on the best of those two.

**Methods And Evaluation Criteria:**

Methods
Modeling Framework: The authors adopt a strategic classification framework in which each applicant’s resume is a feature vector split into “fundamental” (e.g. actual skills/experience) and “style” features (e.g. grammar, resume organization). The LLM stochastically rewrites style features but does not alter fundamental features.
Scoring Function: The hiring side uses a fixed, off-the-shelf resume “scorer” (like an applicant-tracking system) to assign numeric relevance scores. The classifier decides on a threshold: those with scores above the threshold receive a positive decision.
No False Positives Objective: To reflect the practical cost of hiring unqualified candidates, they primarily study the setting where the hiring system tries to maintain zero FPR and maximize TPR.

Evaluation Criteria
1. True Positive Rate (TPR) – fraction of truly qualified applicants who pass the threshold.
2. False Positive Rate (FPR) – fraction of unqualified applicants who pass the threshold (aiming for zero).
3. Fairness Metric: TPR disparity across “privileged” vs. “unprivileged” LLM-access groups.

**Other Comments Or Suggestions:**

None.

**Other Strengths And Weaknesses:**

Strengths
1. As generative AI becomes more common in hiring processes, this is among the first formal frameworks analyzing LLM-based resume enhancements.
2. The theorems clearly demonstrate why (and when) two-ticket classification can reduce disparities without sacrificing accuracy.
3. Empirical tests on genuine resumes using an open-source application-tracking system add credibility to the arguments.

Weaknesses
1. I am concerned about the novelty of this paper.
(1) Using one LLM model for different job roles may limit real-world applicability, as each role’s requirements can differ.
(2) The sample size of 520 resumes might be too small to capture broader labor market heterogeneity.
2. This work relies on generic APIs. Results rely on off-the-shelf LLMs with no domain-specific training. This raises questions about reproducibility if LLMs or prompts change. Future LLM updates or new models could alter outcomes significantly.
3. They rely on a fixed resume-scoring model and do not deeply test alternative commercial ATS or more complex ML-based hiring systems. Real hiring pipelines may also incorporate more than a single numeric score.
4. They model just two groups (privileged vs. unprivileged). Real-world disparities can be multi-faceted.
5. This work focuses on TPR under No-False-Positives. This can make sense in certain scenarios but may not align with all employers’ priorities (some might tolerate a small FPR for a greater TPR). The paper’s approach is specialized, and a user wanting a more flexible trade-off might need a different theoretical framework.

**Questions For Authors:**

None.

**Relation To Broader Scientific Literature:**

1. The paper’s strategic classification framework extends Hardt et al. (2016) and subsequent works on manipulations that alter one’s “input” to a classifier. However, unlike classical strategic classification where manipulations often incur a cost, the authors note that cost here is nearly zero—anyone can prompt an LLM.
2. The fairness approach (equalizing TPR or at least reducing disparity) connects to the well-known “equalized odds” concept, but it emphasizes a special case: no false positives, maximizing TPR.

**Theoretical Claims:**

1. Inequity Under Traditional Schemes: If privileged and unprivileged groups have different-quality LLMs (the privileged group’s LLM stochastically dominates the unprivileged group’s), the TPR of the privileged group will be higher. (Formalized in Theorem 1 and Corollary 1.)

2. Improvement via Two-Ticket Scheme: By giving everyone an additional LLM pass from a strong model, the paper proves that:
a. TPR disparity can only decrease.
b. TPR for both groups can only increase (or remain the same).
c. Overall accuracy remains the same or improves (since TPR rises at zero FPR).

3. Constant Threshold: Under mild assumptions (e.g., the hiring LLM does not exceed the privileged group’s LLM in quality), the threshold that satisfies a zero-FPR constraint does not change when moving from traditional to two-ticket schemes—so gains in fairness/TPR occur “automatically” without requiring re-tuning for each group.

---

> ### Author Rebuttal · Authors · 2025-03-30
>
> We thank the reviewer for the detailed comments and feedback on our work. In particular, we appreciate the suggestion to connect the work to human-in-the-loop hiring and “AI detection” strategies, and will discuss them in the next version of the paper.
>
> We want to highlight that our main contributions are (a) adjusting the strategic classification model for LLM manipulations, and (b) suggesting a first generic solution with theoretical guarantees for this problem. The novelty of our paper comes from the unique analysis of the system effects of some candidates deciding to use (more advanced) LLM tools for their resume. The experiments are added to demonstrate the applicability of the results as empirical computer science experiments. We leave a broader study of this effect across the labor market to future work by domain experts. We would like to provide additional clarifications for the weaknesses listed:
> 1. **Novelty Concerns**: (1) Using one LLM for all roles would be limiting; however, we test the manipulation of different LLMs on resumes of different roles available in the open-source dataset that we use. We are not entirely sure if this was the question the reviewer intended to ask, but we are happy to clarify and answer the question with additional clarification.
> (2) Since the goal of our experiments is to validate our theoretical results, our experiments focus on a few job descriptions and consider only a subset of a diverse labor market (We have added additional experience for 8 more job descriptions). We believe that for our purposes, 520 real resumes are enough.
> 2. **Reproducibility**: The reviewer makes a great point in highlighting the importance of replicability. We have strongly prioritized replicability by using only open-source resumes, an open-source resume scorer, and listing the specific API version of the models we use. While new models might be introduced, our results can always be replicated with the model versions we report. Our work on general-purpose LLMs is motivated by the resume prompts in real-world datasets such as WildChat. Domain-specific training for resume editing is an interesting suggestion, but out of scope of our current work. However, it is important to note that our model and theoretical guarantees are general enough to be applicable to privileged groups accessing domain-specific tools as well.
> 3. **Real Hiring Pipelines**: We agree with the reviewer that real hiring pipelines may incorporate more than a single score. Indeed, local hiring laws such as NYC144 incentivize humans to be in the loop for hiring decisions in order to avoid algorithmic auditing ([source](https://www.fisherphillips.com/en/news-insights/new-york-lawmakers-aim-to-close-loopholes-nycs-ai-bias-audit-law.html)). We want to clarify that we are focused on the first stage of the hiring process: resume filtering. This is a necessary step candidates must pass and is done primarily by ATS software such as the one we test. We focus on the only available open-source ATS scoring system (to the best of our knowledge) in order to ensure reproducibility of our results. Moreover, our model of the resume scoring system is general and can incorporate and combine multiple metrics used to make a filtering decision: we only require that the system is monotonic in the resume features. We will make sure to address that in the next version of the paper.
> 4. **Multifaceted “privileged” groups**: While we address only two levels of privileged groups, all our results can be easily generalized to multiple levels of privilege, so long as the Hirer uses a Pareto-dominant LLM (that is at least as good as the most privileged group’s LLM): this follows from how we define LLM manipulations stochastically. We will make sure to address this in the next version of the paper! In case the reviewer means other types of disparities, our protected attribute that we introduce and are interested in is model-access privilege.
> 5. **Minimizing False Positive Rates:** Our research provides an initial step into utilizing LLMs to reduce inequities in hiring procedures as a result of LLM manipulations, with theoretical guarantees on improvements with the Two-Ticket scheme. Our research is most relevant to job positions receiving large numbers of applications. Similarly to related works, we find that there is sufficient evidence to believe that a FPR=0 constraint is necessary when there is a large volume of applicants for a select number of positions, as per the standard in hiring for tech sector positions nowadays. For example, one mid-size company may receive over 25,000 applications for just 6 summer intern positions in 2024 ([source](https://www.linkedin.com/posts/patreon_patreonintern-launchingwithpatreon-internprograms-activity-7067615159833280512-A42d/)).

---

### Official Review · Reviewer_rGk7 · 2025-03-14

**Overall Recommendation:** 3

**Summary:**

* This paper proposes and investigates a theoretical model for LLM strategic manipulations in the job application market, motivated by empirical observations.
* The model is motivated by three empirical observations: (1) LLMs tend to improve the score of a resume in an automated ranking system, (2) higher-cost LLMs tend to induce a larger improvement in the resume score, and (3) repeated application of LLM improvement steps yields diminishing improvements to the resume score.
* In the formal model, each candidate is represented by a triplet $(x,g,y)$. The feature vector $x\\in\\mathbb{R}^d$ is the candidate’s original resume, represented as a concatenation of immutable (fundamental) and style features. The group $g$ (privileged/unprivileged) represents the candidate’s manipulation ability, and $y$ is a binary label indicating the true qualification level of the candidate. It is assumed that $x$ is independent of $g$, and that the label is independent of group membership.
* LLM manipulation $L$ is modeled as a stochastic function which replaces some of the features in the resume with independent samples from random variables. Hiring decisions are represented by a non-decreasing score function $s(x)$, and it is assumed that each group has access to its own LLM $L_g$, and that the hiring agent also has access to an LLM.
* The goal of the hiring is to maximize the hiring rate of qualified candidates, under the constraint that no unqualified agents are hired. The goal of the candidate is to maximize their probability of acceptance, and they can choose whether to manipulate their resume using their group’s LLM.
* In the theoretical analysis, the authors show that hiring schemes that ignore manipulation induce group disparity. In response, the authors propose the two-ticket scheme, in which the hiring agent may apply additional LLM manipulation on top of the submitted application, and show that the two-ticket scheme decreases disparity under assumptions.
* Finally, the empirical section simulates the process using real resumes and using OpenAI LLMs for manipulation, showing favorable results.

## Update after rebuttal
Thank you again for the response. The rebuttal addressed some of my concerns regarding the validity of the assumptions (in particular, I agree that it is reasonable to assume that monthly LLM subscription is more accessible to privileged groups), while I still believe that other foundational assumptions would greatly benefit from stronger grounding or refinement (in particular, the formal model of LLM manipulation). With this, I maintain my original assessment, and I believe that the paper is a step is the right direction.

**Claims And Evidence:**

The paper’s theoretical claims and empirical evaluations seen to be well-supported. Connection to practical problems relies on very strong assumptions, and the degree to which this disparity may appear in practical scenarios is not completely clear:
* Resumes typically contain thousands of tokens, and the current LLM prices for refining documents with thousands of tokens are currently in the order of cents, even for the most expensive LLMs - A relatively negligible amount.
* LLMs are often used interactively, with multiple revision steps that might not align with the single-shot manipulation modeled here.
* The model relies on the assumption that “style features” can be decoupled from “fundamental features” - However it is not clear to what degree this assumption holds in practice. For example, product management and UX design (the jobs under consideration in the empirical evaluation section) rely on communication skill, where the ability to communicate well in written form is a key requirement. It is therefore unclear whether it is reasonable to decouple “fundamental” and “style” features in this case.
* While in practice it seems very reasonable to assume that LLM proficiency varies across groups, it’s not clear whether the conditional independence assumptions on $x$ and $y$ are likely to hold in this case.

**Essential References Not Discussed:**

It may be valuable to acknowledge related work that intersects strategic classification and labor market analyses, such as Somerstep et al. (“Learning In Reverse Causal Strategic Environments With Ramifications on Two Sided Markets”, ICLR 2024).

**Experimental Designs Or Analyses:**

* The overall experimental design is reasonable.
* The reports results on 520 CVs, the supplementary material only includes revised samples from 100 CVs. This discrepancy should be clarified.

**Methods And Evaluation Criteria:**

The proposed evaluation method appears sound.

**Other Comments Or Suggestions:**

Some of the notations in Section 4 were confusing -  L199 says that “the formulation is based on the observation that LLMs can standardize style features…”, however Definition 4.1 formally defines the LLM manipulation $L$ as a transformation which modifies the values of the “fundamental features” $x_i$ (defined around L133), which seems to contradict this.

**Other Strengths And Weaknesses:**

Strengths:
* Clean theoretical model, which relies on a simple and natural empirical observation.
* Theoretical claims are supported by empirical observations.
* Paper is very clear and easy to follow.
Weaknesses:
* Model relies on strong assumptions, and it is unclear whether the described disparity is likely to be significant in practical scenarios.

**Questions For Authors:**

* Is it possible to describe a practical scenario where access to LLMs is likely to be the significant factor in decision making, while the independence/conditional independence assumptions still hold? (i.e. a scenario $x$ and $y$ are independent/conditionally independent of the group as assumed, and there is a difference in LLM accessibility for resume improvements)
* How would the model and its results change if candidates were allowed to interact iteratively with the LLM rather than performing a single-shot manipulation?
* What happens if one LLM does not Pareto-dominate another? (i.e., when the conditions in Definition 5.3 do not hold)

**Relation To Broader Scientific Literature:**

* The paper provides a practical classification for strategic classification, motivated by LLMs. The traditional strategic classification literature is focused on classic statistical learning problems (classification/regression), and modeling strategic behavior in the context of generative AI is an emerging research frontier.
* Not clear if the proposed model is applicable to other scenarios beyond labor markets.

**Theoretical Claims:**

The theoretical derivations appear sound at a high level. A deeper verification of the proofs can confirm the robustness of these claims.

---

> ### Author Rebuttal · Authors · 2025-03-30
>
> We appreciate the reviewer’s detailed comments and feedback on our work. Here we will focus on answering questions from the reviewer:
> 1. **Practical scenario of X and Y independence**: Just to clarify our assumptions, our paper builds on existing fairness literature and assumes that “X and G are independent, and Y and G are conditionally independent given X.” One example with these assumptions is non-native English speakers who are equally qualified for a project management job, but the applicant who uses the better LLM might achieve a higher scoring resume in the system. We are also happy to clarify further if we didn’t fully answer this question.
> 2. **Iterative interaction with LLMs**: While LLMs can indeed be used interactively with multiple revisions, our single-shot manipulation model provides a foundational understanding of strategic classification in the era of LLMs. For example, we observed significant improvements for "style features’’ under one application of LLM manipulation. As LLMs improve, we find these multiple revision steps are less helpful. That said, future work can explore extending the model to incorporate iterative, interactive revisions, potentially offering even finer-grained control and adaptability in practical scenarios.
>
> To address iterative manipulations on the Candidate's side, we can assume the distribution of manipulated features already captures this. On the Hirer’s side, iterative manipulations would be more challenging in practice as it will require them to have a person in charge of such interactions, which might not be ideal considering that the candidate knows their background better than anyone else. However, we think such interactions will have diminishing returns of improvement, and that having access to more advanced LLMs is more crucial. We will add this discussion in the next version of the paper.
>
> 3. **No Pareto-dominance**: Great question! When there are free vs. premium versions of an LLM, we find Pareto dominance is a reasonable assumption. Without Pareto-dominance, group outcomes depend highly on the scoring system deployed, and the two-ticket scheme bias mitigation could be reduced. For example, if Group P’s LLM improves feature 1 and Group U’s LLM improves feature 2, and the Hirer uses the same LLM as Group U, it could be that the screening software gives no weight to feature 2. An interesting direction for future work could be to apply multiple different LLMs to improve a single resume, both from the Candidate’s side and the Hirer’s side. We will clarify this in the next version of the paper.
>
> We will offer several clarifications to some points raised around broader applicability.
> - **Token-based costs**: We agree that token-based costs for refining long resumes are minimal. However, many candidates are unaware/don’t know how they pay per token and simply default to whichever premium LLM they can access, which can widen disparities if others rely on free or older models. Our motivation also addresses cases in which specialized, high-cost resume-editing tools can be used by some people (We briefly discuss this assumption in Section 4.2).
> - **“Fundamental” and “style” features**: The decoupling of "style features" from "fundamental features" in our model is for analytical purposes to demonstrate that LLMs can enhance the style of resumes without altering their core qualifications. This does not imply that style features are unimportant or that evaluation scores do not depend on them. Moreover, if every feature of relevance is a style feature, then arguably LLMs make resumes a bad indicator of qualifications for those jobs (in which case LLM detection methods could be relevant).
> - **Experiments in the supplementary material**: We used 100 sampled CVs in the supplementary material due to budget restrictions. We are happy to expand the experiments to more resumes.
> - **Applicability**: Beyond the labor market, our work is applicable in any application system, e.g., college and graduate school applications, where a judge’s perception of quality is based on features that an LLM can improve (without fabricating lies).
> - **Somerstep et al.**: Great suggestion! We specifically note the relevance of this work, which explores causal strategic classification and its impact on labor market dynamics. This line of research complements our work by providing insights into how strategic behavior can influence both employer and labor force outcomes, highlighting the importance of considering these dynamics in developing fair and effective hiring practices. We will ensure to incorporate this discussion in the paper.
> - **Section 4 Notation**: Thank you for noticing this! We assume LLMs change only style features. We will correct this.
>
> We thank the reviewer for the detailed reading of our work. We hope that our response provided clarification and strengthened our contribution. Please let us know if we can answer any additional questions.

---

> > ### Comment · Reviewer_rGk7 · 2025-04-05
> >
> > Thank you for the detailed response! It clarifies some of my concerns, and I maintain my original score.

---

### Official Review · Reviewer_vS3u · 2025-03-23

**Overall Recommendation:** 3

**Summary:**

The paper explores challenges of fairness and accuracy in hiring when job seekers use generative AI tools to enhance resumes. It proposes a "two-ticket" scheme, where employers also manipulate resumes using AI. The study demonstrates, theoretically and empirically, that this approach improves fairness and accuracy in the recruitment process.

**Claims And Evidence:**

The claims made in the paper are well-supported by both theoretical reasoning and empirical evidence. However, broader validation across a more extensive range of job roles and industries could strengthen the generalizability of their findings. Additionally, more details about the potential variability in effectiveness with different types of jobs and applicant tracking systems could further validate the claims

**Essential References Not Discussed:**

None

**Experimental Designs Or Analyses:**

Yes

**Methods And Evaluation Criteria:**

The proposed methods and evaluation criteria are appropriate for addressing fairness and accuracy challenges in hiring when candidates use generative AI tools. The "two-ticket" scheme and the use of real-world resume datasets effectively demonstrate improvements in these areas under the study's framework.

**Other Comments Or Suggestions:**

None

**Other Strengths And Weaknesses:**

None

**Questions For Authors:**

What technical or logistical challenges might arise in implementing the “two-ticket” scheme across various industries, and how can these challenges be addressed?

What measures are in place to ensure the protection of candidates’ data privacy when resumes are manipulated using generative AI, and how can ethical concerns be mitigated?

How might the “two-ticket” scheme affect employee performance and satisfaction in the long run, and what plans are there for further research into its long-term impact and adaptability across different job sectors?

**Relation To Broader Scientific Literature:**

The paper addresses fairness and accuracy issues in algorithmic hiring processes influenced by generative AI tools that manipulate resumes. It proposes the "two-ticket" scheme to mitigate bias. The study builds on existing research in algorithmic bias, strategic classification, and AI's impact on resume quality, offering new insights to balance fairness and accuracy.

**Theoretical Claims:**

Yes，the paper provides detailed mathematical formulations and accompanying proofs aimed at demonstrating the validity of the “two-ticket” scheme and its impact on fairness and accuracy in hiring processes. Any issues would require direct examination and verification of these proofs by an expert in the field.

---

> ### Author Rebuttal · Authors · 2025-03-30
>
> We thank the reviewer for the thoughtful summary and comments on our paper. We address the questions below:
> 1. **Technical/Logistical Challenges**: In the real world, there may be practical challenges in implementing our “two-ticket” scheme. A major challenge is for companies to decide which model to use when applying the second manipulation. Companies will likely test different models to determine which one best improves resume scores in their ATS system, based on their specific job description style, before selecting a model for the second manipulation. Not all companies may have the technical expertise to do this kind of model selection. We will address this in the next version of the paper.
> 2. **Protection of Data Privacy**: This is an excellent question. We assume that companies use APIs in a way that queries are not stored by the company (this is the case, for example, for the LLMs accessible to people from the university the authors are affiliated with). Different companies offer ways to opt out of data collection. We do not anticipate the resume manipulation itself to be any less private than the companies storing a candidate's resume. Another way that privacy might be compromised is from the chosen threshold. The threshold might be sensitive to the resume score of a single individual, violating exact differential privacy. For the threshold to be privacy-preserving, we recommend using DP threshold functions [1]. We will make sure to discuss this in the next version of the paper.
> 3. **Long-term / Downstream Impacts**: Considering the downstream impacts is a fascinating perspective on the system effects of LLM tools used for job applications. In this work, we look at the applicant filtering stage of the hiring process. The subsequent hiring decisions made by humans may introduce further disparities that may affect employee satisfaction in the long run. In future work, we plan to do an empirical study of the impact of LLM-aided application materials across a variety of industries in consultation with economists.
>
> We thank the reviewer for providing insightful and thought-provoking questions about the practical implications of our work. We hope our answers help address the reviewer’s concerns and strengthen the contributions of our paper. Our main contribution is to provide an intuitive theoretical framework for modeling LLM manipulations in the applicant screening process. We are happy to answer any additional questions.
>
> References:
> [1] Bun, M., Nissim, K., Stemmer, U., & Vadhan, S. (2015, October). Differentially private release and learning of threshold functions. In 2015 IEEE 56th Annual Symposium on Foundations of Computer Science (pp. 634-649). IEEE.

---

> > ### Comment · Reviewer_vS3u · 2025-04-01
> >
> > The author's rebuttal effectively reduce my concerns, and I maintained my score

---

### Official Review · Reviewer_BLGs · 2025-03-25

**Overall Recommendation:** 4

**Summary:**

The paper considers the problem of hiring in the scenario when applicants use LLMs to assist in CV writing (as well as hirers can have their own LLMs). This potentially can lead to unfair and inaccurate hiring, if, say some applicants use paid version of LLM while others do not. To mitigate this, the authors propose a two-ticket scheme where the hiring algorithm considers both the original CV and a manipulated version of it. They provide theoretical proof that this scheme improves fairness and accuracy in hiring when TPR is maximized s.t. FPR = 0.

**Claims And Evidence:**

Claim 1: The paper establishes that hiring can be unfair when different applicants use different LLMs -- Figure 1.

Claim 2: Two-ticket scheme is introduced and proved that it improves disparity -- In two-ticket scheme the hiring algorithm considers both the original and a manipulated version of each resume. Theoretical improvements in fairness and accuracy are demonstrated through Theorem 2 and Corollary 2, which works under No False Positives Objective.

Claim 3: The results are empirically validated through a case study in Section 7.

**Essential References Not Discussed:**

NA

**Experimental Designs Or Analyses:**

Appendix C provides more details on evaluation and seems reasonable.

**Methods And Evaluation Criteria:**

Yes, different LLMs are considered, uncertainty intervals are reported.

**Other Comments Or Suggestions:**

NA

**Other Strengths And Weaknesses:**

The paper is well-written and is easy to read.

Among weaknesses, some of the assumptions of the theoretical analysis might be strict, including no false positives objective.

**Questions For Authors:**

Does the framework hold if manipulations apply to style features?

Accuracy did not change much for the two-ticket scheme compared to traditional evaluation, increasing by around 5%. Do you observe a connection between manipulation strength and accuracy change?

**Relation To Broader Scientific Literature:**

The paper extends the strategic classification framework to address LLM-driven manipulations, which introduce stochasticity and complexity. This paper offers a valuable initial step towards fairness analysis in LLM-based hiring scenarios. The claims are supported both theoretically and empirically.

**Theoretical Claims:**

I read Appendix E and theoretical results in the main paper - they appear correct to me.

---

> ### Author Rebuttal · Authors · 2025-03-30
>
> We would like to thank the reviewer for the comments and for reading our paper (even the appendix!). We will answer the questions below:
> > Does the framework hold if manipulations apply to style features?
>
> Our framework considers manipulations made to style features rather than fundamental features. We accidentally referred to style features as fundamental when we introduced them. The typo occurs in lines 134 - 137; style features should be $(x_1, …, x_{d_1})$ while fundamental features should be $(c_1, …, c_{d_2})$. This notation is consistent in the rest of our paper. In this framework, style features are manipulable through LLM manipulations while fundamental features such as technical experience and programming skills are preserved. We thank the reviewer for catching this key typo.
>
> > Accuracy did not change much for the two-ticket scheme compared to traditional evaluation, increasing by around 5%. Do you observe a connection between manipulation strength and accuracy change?
>
> Regarding accuracy change, we expect that stronger manipulations will lead to a greater accuracy improvement in our two-ticket scheme. For example, we expect to see the highest improvement in accuracy when qualified and unqualified resumes are separable originally and become more difficult to separate after a round of candidate manipulation. In Figure 1, we see that for PM resumes, this effect was more pronounced for Claude-3.5-Sonnet, GPT4o, and Llama3-70B. In contrast, when there is not much change in scores, the accuracy remains the same for both traditional and two-ticket schemes. We will make sure to discuss this in the next version of the paper!
>
> We are happy to answer any additional questions! We thank the reviewer for understanding and appreciating our non-traditional contribution.

---

> > ### Comment · Reviewer_BLGs · 2025-04-01
> >
> > Thank you for more details. I have read other reviews and responses and will keep my score as is.

---

### Decision · Program_Chairs · 2025-05-01

**Decision:**

Accept (poster)

**Comment:**

The paper studies a variant of strategic classification that corresponds to automated hiring when applicants use LLMs to improve their CVs.  The reviewers all agreed that the paper is a valuable contribution, and the main (and relatively minor) complaint was about strong assumptions.  We encourage the authors to improve the paper accordingly based on the detailed suggestions and comments in the reviews.